# MDAgents: An Adaptive Collaboration of LLMs for Medical Decision-Making

**Yubin Kim**[1]    **Chanwoo Park**[1]    **Hyewon Jeong**[1♮]    **Yik Siu Chan**[1]
**Xuhai Xu**[1] **Daniel McDuff**[2]    **Hyeonhoon Lee**[3]
**Marzyeh Ghassemi**[1]    **Cynthia Breazeal**[1]    **Hae Won Park**[1]
[1]Massachusetts Institute of Technology
[2]Google Research
[3]Seoul National University Hospital
{ybkim95,cpark97,hyewonj,yiksiuc,xoxu,mghassem,cynthiab,haewon}@mit.edu
dmcduff@google.com
hhoon@snu.ac.kr

## Abstract

Foundation models are becoming valuable tools in medicine. Yet despite their promise, the best way to leverage Large Language Models (LLMs) in complex medical tasks remains an open question. We introduce a novel multi-agent framework, named **M**edical **D**ecision-making **Agents** (**MDAgents**) that helps to address this gap by automatically assigning a collaboration structure to a team of LLMs. The assigned solo or group collaboration structure is tailored to the medical task at hand, a simple emulation inspired by the way real-world medical decision-making processes are adapted to tasks of different complexities. We evaluate our framework and baseline methods using state-of-the-art LLMs across a suite of real-world medical knowledge and medical diagnosis benchmarks, including a comparison of LLMs' medical complexity classification against human physicians[2]. MDAgents achieved the **best performance in seven out of ten** benchmarks on tasks requiring an understanding of medical knowledge and multi-modal reasoning, showing a significant **improvement of up to 4.2%** ($p < 0.05$) compared to previous methods' best performances. Ablation studies reveal that MDAgents effectively determines medical complexity to optimize for efficiency and accuracy across diverse medical tasks. Notably, the combination of moderator review and external medical knowledge in group collaboration resulted in an average accuracy **improvement of 11.8%**. Our code can be found at https://github.com/mitmedialab/MDAgents.

## 1   Introduction

Medical Decision-Making (MDM) is a multifaceted and intricate process in which clinicians collaboratively navigate diverse sources of information to reach a precise and specific conclusion [97]. For instance, a primary care physician (PCP) may refer a patient to a specialist when faced with a complex case, or a patient visiting the emergency department or urgent care might be triaged and then directed to a specialist for further evaluation [5, 54]. MDM involves interpreting complex and multi-modal data, such as imaging, electronic health records (EHR), physiological signals, and genetic information, while rapidly integrating new medical research into clinical practice [68, 78]. Recently, Large Language Models (LLMs) have shown potential for AI support in MDM [22, 37, 48, 64, 72, 90]. It is known that they are able to process and synthesize large volumes of medical literature [74] and clinical information [1], as well as support probabilistic [94] and causal [39] reasoning, makes LLMs promising tools. However, there is no silver bullet in medical applications that require careful design.

---

♮Hyewon Jeong received her MD degree from Yonsei University College of Medicine, South Korea.

[2]Appendix F contains a detailed comparison results between human physicians and LLMs.

38th Conference on Neural Information Processing Systems (NeurIPS 2024).

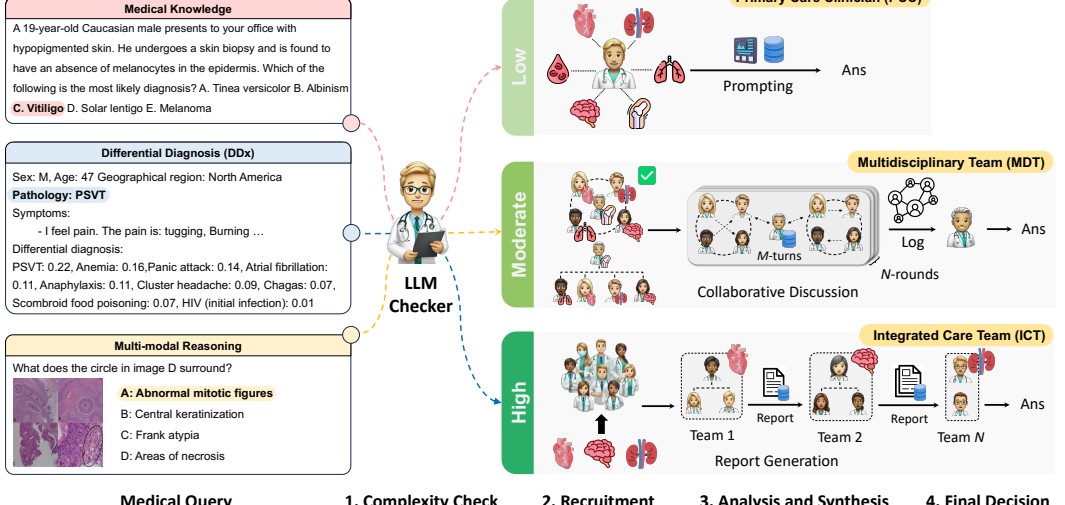

Figure 1: **Medical Decision-Making Agents (MDAgents) framework**. Given a medical query from different medical datasets, the framework performs 1) medical complexity check, 2) recruitment, 3) analysis and synthesis, and 4) decision-making steps.

While decision-making tools including multi-agent LLMs [11, 86] have shown promise in non-medical domains [31, 32, 44, 46, 62, 65], their evaluation in health applications has been limited. To date, their "generalist" design has not effectively integrated the real-world systematic MDM process [57] which requires an adaptive, collaborative, and tiered approach. Clinicians consider the current and past history of the patient, available evidence from medical literature, and their domain expertise and experience [20] for MDM. One example of MDM is to triage patients in emergency room based on the severity and complexity of their medical conditions [12, 26, 87]. Patients with pathognomonic, single uncomplicated acute conditions, or stable chronic conditions that their PCP could manage [85] could be low complexity cases. On the other hand, patients with injuries that involve multiple organs, chronic conditions with side effects, or superimposed diseases who often require multiple collaborative discussions (MDT) or sequential consultations (ICT) among specialty physicians [27, 61] are considered high complexity cases [3].

Inspired by the way that clinicians make decisions in practice, we propose **M**edical **D**ecision-making **Agents** (**MDAgents**), an adaptive medical decision-making framework that leverages LLMs to emulate the hierarchical diagnosis procedures ranging from individual clinicians to collaborative clinician teams (Figure 1). MDAgents work in three steps: 1) Medical complexity check; 2) Recruitment based on medical complexity; 3) Analysis and synthesis and 4) Final decision-making to return the answer. Our contributions are threefold:

1. We introduce MDAgents, the first adaptive decision-making framework for LLMs that mirrors real-world MDM processes via dynamic collaboration among AI agents based on task complexity.
2. MDAgents demonstrate superior performance in accuracy over previous solo and group methods on 7 out of 10 medical benchmarks, and we show an effective trade-off between performance and efficiency (i.e. the number of API calls) by varying the number of agents.
3. We provide rigorous testing under various hyperparameters (e.g. temperatures), demonstrating better robustness of MDAgents compared to solo and group methods. Furthermore, our ablations evidence MDAgents' ability to find the appropriate complexity level for each MDM instance.

## 2   Related Works

**Language Models in Medical Decision-Making**   LLMs have shown promise in a range of applications within the medical field [14, 37, 40, 48, 63, 75, 76, 90, 96]. They can answer questions from medical exams [43, 52], perform biomedical research [36], clinical risk prediction [37], and clinical diagnosis [55, 67]. Medical LLMs are also evaluated on generative tasks, including creating medical reports [79], describing medical images [81], constructing differentials [53], performing diagnostic dialogue with patients [77], and generating psychiatric evaluations of interviews [24]. To advance the capabilities of medical LLMs, two main approaches have been explored: (1) training with

---

[3]Detailed examples of low-, moderate- and high-complexity cases are provided in Appendix E.1.

Table 1: Comparison between our framework and previous methods (Solo and Group). Among these works, MDAgents is the only one to perform all key dimensions of LLM decision-making.

| Method | MDAgents (Ours) | Single | Voting [82] | Debate [17] | MedAgents [72] | ReConcile [10] |
|---|---|---|---|---|---|---|
| Interaction Type | | | | | | |
| Multiple Roles | ✓ | ✗ | ✓ | ✓ | ✓ | ✓ |
| Early Stopping | ✓ | ✗ | ✓ | ✓ | ✓ | ✗ |
| Refinement | ✓ | ✗ | ✗ | ✓ | ✓ | ✗ |
| Complexity Check | ✓ | ✗ | ✗ | ✗ | ✗ | ✗ |
| Multi-party Chat | ✓ | ✗ | ✗ | ✓ | ✗ | ✗ |
| Conversation Pattern | Flexible | Static | Static | Static | Static | Static |

domain-specific data [28], and (2) applying inference-time techniques such as prompt engineering [67] and Retrieval Augmented Generation (RAG) [92]. While initial research has been concentrated on pre-training and fine-tuning with medical knowledge, the rise of large general-purpose LLMs has enabled training-free methods where models leverage their latent medical knowledge. For example, GPT-4 [60], with richer prompt crafting, surpasses the passing score on USMLE by over 20 points and with prompt tuning can outperform fine-tuned models including Med-PaLM [58, 59]. The promise of general-purpose models has thus inspired various techniques such as Medprompt and ensemble refinement to improve LLM reasoning [67], as well as RAG tools that use external resources to improve the factuality and completeness of LLM responses [38, 92]. Frameworks like MEDIQ [49] and UoT [33] advance LLM reliability in clinical settings by enhancing information-seeking through adaptive question-asking and uncertainty reduction, supporting more realistic diagnostic processes. Our approach leverages these techniques and the capabilities of general-purpose models while acknowledging that a solitary LLM [37, 48, 90] may not fully encapsulate the collaborative and multidisciplinary nature of real-world MDM. We thus emphasize joinging multiple expert LLMs for effective collaboration in order to solve complicated medical tasks with greater accuracy.

**Multi-Agent Collaboration**  An array of studies have explored effective collaboration frameworks between multiple LLM agents [47, 86] to enhance capability above and beyond an individual LLM [80]. A common framework is role-playing, where each agent adopts a specific role (e.g. an Assistant Agent or a Manager Agent), a task is then broken down into sub-steps and solved collaboratively [47, 86]. While role-playing focuses on collaboration and multi-step problem-solving [88], another framework, "multi-agent debate", prompts each agent to solve the task independently [17]. Then, they reason through other agents' answers to converge on a shared response, this approach can improve the factuality, mathematical ability and reasoning capabilities of the multi-agent solution [17, 50]. Similar frameworks include voting [82], multi-disciplinary collaboration [72], group discussions (ReConcile [10]), and negotiating [23]. Table 1 compares existing setups across key dimensions in multi-agent interaction. Although these frameworks have shown improvement in the respective tasks, they rely on a pre-determined number of agents and interaction settings. When applied on a wider variety of tasks , this static architecture may lead to suboptimal multi-agent configurations, negatively impacting performance [51]. Furthermore, multi-agent approaches run the risk of being computationally inefficient or expensive to employ and need to justify these costs with noticable performance gains [17]. Given that different models and frameworks could generalize better to different tasks [93], we propose a framework that dynamically assigns the optimal collaboration strategy at inference time based on the complexity of the query. We apply our strategy to MDM, a task that requires teamwork and should benefit from multi-agent collaboration [72].

## 3   MDAgents: Medical Decision-making Agents

The design of MDAgents (Figures 1 and 2) incorporates four stages: **1) Medical Complexity Check** - The system evaluates the medical query, categorizing it as *low*, *moderate*, or *high* complexity based on clinical decision-making techniques [6, 7, 21, 71, 84]. **2) Expert Recruitment** - Based on complexity, the framework activates a single Primary Care Clinician (PCC) for low complexity issues, or a Multi-disciplinary Team (MDT) or Integrated Care Team (ICT) for moderate or high complexities [7, 18, 21, 34, 45, 71]. **3) Analysis and Synthesis** - Solo queries use prompting techniques like Chain-of-Thought (CoT) and Self-Consistency (SC). MDTs involve multiple LLM agents forming a consensus, while ICTs synthesize information for the most complex cases. **4) Decision-making** - The final stage synthesizes all inputs to provide a well-informed answer to the medical query.

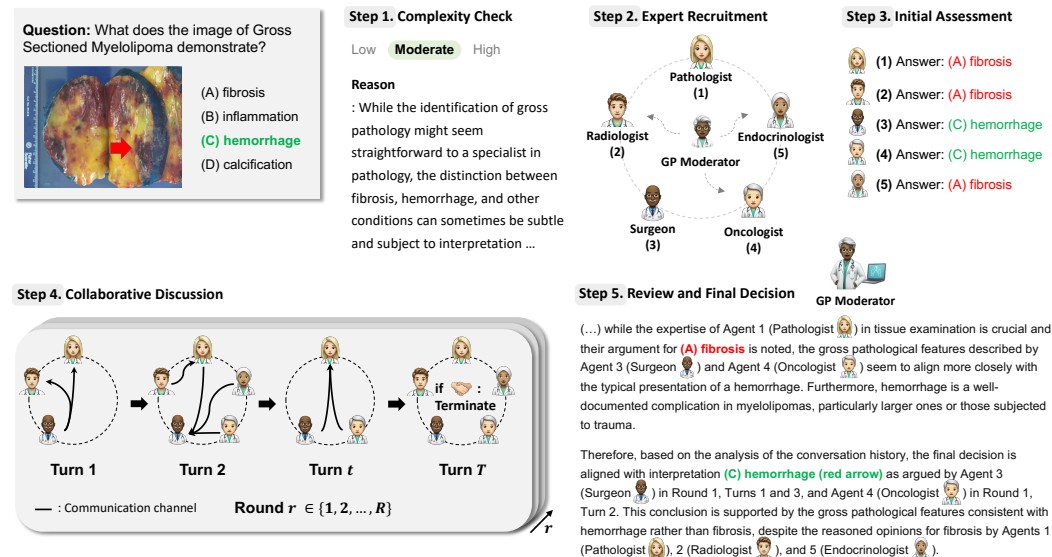

Figure 2: Illustrative example of MDAgents in a *moderate* complexity case from the PMC-VQA dataset. More detailed case studies can be found in Figure 11 and 12 in the Appendix.

## 3.1 Agent Roles

**Moderator.** The moderator agent functions as a general practitioner (GP) or emergency department doctor who first triages the medical query. This agent assesses the complexity of the problem and determines whether it should be handled by a single agent, a MDT, or an ICT. The moderator ensures the appropriate pathway be selected based on the query's complexity and oversees the entire decision-making process.

**Recruiter.** The recruiter agent is responsible for assembling the appropriate team of specialist agents based on the complexity assessment of the moderator. The recruiter may assign a single PCP agent for low-complexity cases, while MDT or ICT with relevant expertise will be formed for moderate and high-complexity cases.

**General Doctor/Specialist.** These agents are domain-specific or general physicians recruited by the recruiter agent. Depending on the complexity of the case, they may work independently or as part of a team. General physicians handle less complex, routine cases, whereas specialists are recruited for their specific expertise in more complex scenarios. These agents engage in the collaborative decision-making process, contributing their specialized knowledge to reach a consensus or provide detailed reports for high-complexity cases.

## 3.2 Medical Complexity Classification (Line 1 of Algorithm 1, Appendix C.2)

The first step in the MDAgents framework is to determine the complexity of a given medical query $q$ by the *moderator* LLM which functions as a *generalist practitioner* (GP). The moderator aims to act as a classifier to return the complexity level of the given medical query, it is provided with the information on how medical complexity should be defined and is instructed to classify the query into one of three different complexity levels:

1. *Low* - Simple, well-defined medical issues that can be resolved by a single PCP agent. These typically include common, acute illnesses or stable chronic conditions where the medical needs are predictable and require minimal interdisciplinary coordination.
2. *Moderate* - The medical issues involve multiple interacting factors, necessitating a collaborative approach among an MDT. These scenarios require integration of diverse medical knowledge areas and coordination between specialists through consultation to develop effective care strategies.
3. *High* - Complex medical scenarios that demand extensive coordination and combined expertise from an ICT. These cases often involve multiple chronic conditions, complicated surgical or trauma cases, and decision-makings that integrates specialists from different healthcare departments.

## 3.3 Expert Recruitment (Line 3, 7, 17 of Algorithm 1)

Given a medical query, the goal of the *recruiter* LLM is to enlist domain experts as individuals, in groups, or as multiple teams, based on the complexity levels determined by the *moderator* LLM. Specifically, we assign medical expertise and roles to multiple LLMs, instructing them to either act independently as solo medical agents or collaborate with other medical experts in a team. In Figure 9 in the Appendix, we also provide frequently recruited agents for each benchmark as a reference.

### 3.4 Medical Collaboration and Refinement

The initial assessment protocol of our decision-making framework categorizes query complexity into *low*, *moderate*, and *high*. This categorization is grounded in established medical constructs such as acuity [25] for straightforward cases, comorbidity [69] and case management complexity [13] for intermediate and multi-disciplinary care requirements, and severity of illness [16] for high complexity cases requiring comprehensive management. We outlines the specific refinement approach:

***Low* - Straightforward cases (Line 2-4 of Algorithm 1).**    For queries classified under Low complexity, characterized by straightforward clinical decision pathways, a single PCP agent (Figure 10 (a)) is deployed by the definition in Section 3.2. The domain expert who is recruited by *recruiter* LLM, applies few-shot prompting to the problem. The output answer, denoted as $ans$, is directly obtained from the agent's response to $Q$ without the need for iterative refinement, formalized as $ans = Agent(Q)$, with $Agent$ representing the engaged PCP agent.

***Moderate* - Intermediate complexity cases (Line 6-14 of Algorithm 1).**    In addressing more complex queries, the utilization of an MDT (Figure 10 (b) and (c)) approach has been increasingly recognized for its effectiveness in producing comprehensive and nuanced solutions [45]. The MDT framework leverages the collective expertise of professionals from diverse disciplines, facilitating a holistic examination of the query at hand. This collaborative method is particularly advantageous in scenarios where the complexity of a problem transcends the scope of a single domain, necessitating a fusion of insights from various specialties [7, 71]. The MDT approach not only enhances decision-making quality through the integration of multidimensional perspectives but also significantly improves the adaptability and efficiency of the problem-solving process [21].

Building upon this foundation, our framework specifically addresses queries of moderate complexity through a structured, multi-tiered collaborative approach. An MDT recruited by *recruiter* LLM (see Figure 10 in Appendix) starts an iterative discussion process aimed at reaching a consensus with at most $R$ rounds (Line 10-12). For each round $r \in R$, agents $A_i, i \in 1, \ldots, N$ indicate participation and preferred interlocutors. The system facilitates message exchanges for $T$ turns. If consensus is not reached and agents agree to continue, a new round begins with access to previous conversations. For every round, consensus within the MDT is determined by parsing and comparing their opinions. In the event of a disagreement, the moderator agent, consistent with the one described in Section 3.2 reviews the MDT's discourse and formulates feedback for each agent.

***High* - Complex care cases (Line 17-24 of Algorithm 1).**    In contrast to the MDT approach, the ICT (Figure 10 (d)) paradigm is essential for addressing the highest tier of query complexity in healthcare. This structured progression through the ICT ensures a depth of analysis that is specialized and focused at each stage of the decision-making process. Beginning with the Initial Assessment Team, moving through various diagnostic teams, and culminating with the Final Review & Decision Team, our ICT model aligns specialist insights into a cohesive narrative that informs the ultimate decision (Appendix Algorithms 1 Lines 19-21). A key component of this process is the report generation process described in Appendix with the prompt, where each team, led by a lead clinician, collaboratively produces a comprehensive report synthesizing their findings. This phased approach, supported by evidence from recent healthcare studies, has been shown to enhance the precision of clinical decision-making, as each team builds upon the foundation laid by the previous, ensuring a meticulous and refined examination of complex medical cases [34]. The resultant reports, accumulating throughout the ICT process, are not only reflective of comprehensive medical evaluations but also of a systematic and layered analysis that is critical in the management of intricate health scenarios [18].

### 3.5 Decision-making

In the final stage of our framework, the decision-maker agent synthesizes the diverse inputs generated throughout the decision-making process to arrive at a well-informed final answer to the medical query $q$. This synthesis involves several components depending on the complexity level of the query:

1. *Low*: Directly utilizes the initial response from the primary decision-making agent.

2. *Moderate*: Incorporates the conversation history (*Interaction*) between the recruited agents to understand the nuances and disagreements in their responses.

3. *High*: Considers detailed reports (*Reports*) generated by the agents, which include comprehensive analyses and justifications for their diagnostic suggestions.

The decision-making process is formulated as $ans = Agent(\cdot)$ where the final answer, *ans* is determined by integrating the outputs from analysis and synthesis step based on its medical complexities. This integration employs ensemble techniques such as temperature ensembles to ensure the decision is robust and reflects a consensus among the models when applicable (see Appendix C.2 for details).

Table 2: Accuracy (%) on Medical benchmarks with **Solo/Group/Adaptive** setting. **Bold** represents the best and Underlined represents the second best performance for each benchmark and model. All benchmarks except for MedVidQA were evaluated with GPT-4(V) and MedVidQA was evaluated with Gemini-Pro(Vision). Full experimental results with other models are listed in Table 9-12 in Appendix.

| Category | Method | Medical Knowledge Retrieval Datasets | | | | |
|---|---|---|---|---|---|---|
| | | MedQA | PubMedQA | Path-VQA | PMC-VQA | MedVidQA |
| Single-agent | Zero-shot | 75.0 ± 1.3 | 61.5 ± 2.2 | 57.9 ± 1.6 | 49.0 ± 3.7 | 37.9 ± 8.4 |
| | Few-shot | 72.9 ± 11.4 | 63.1 ± 11.7 | 57.5 ± 4.5 | 52.2 ± 2.0 | 47.1 ± 8.6 |
| | + CoT [83] | 82.5 ± 4.9 | 57.6 ± 9.2 | 58.6 ± 3.1 | 51.3 ± 1.5 | 48.6 ± 5.5 |
| | + CoT-SC [82] | 83.9 ± 2.7 | 58.7 ± 5.0 | 61.2 ± 2.1 | 50.5 ± 5.2 | 49.2 ± 8.2 |
| | ER [67] | 81.9 ± 2.1 | 56.0 ± 7.0 | 61.4 ± 4.1 | 52.7 ± 2.9 | 48.5 ± 4.1 |
| | Medprompt [59] | 82.4 ± 5.1 | 51.8 ± 4.6 | 59.2 ± 5.7 | 53.4 ± 7.9 | 44.5 ± 2.0 |
| Multi-agent (Single-model) | Majority Voting | 80.6 ± 2.9 | 72.2 ± 6.9 | 56.9 ± 19.7 | 36.8 ± 6.7 | 50.8 ± 7.4 |
| | Weighted Voting | 78.8 ± 1.1 | 72.2 ± 6.9 | 62.1 ± 13.9 | 25.4 ± 9.0 | **57.8** ± 2.1 |
| | Borda Count | 70.3 ± 8.5 | 66.9 ± 3.0 | 61.9 ± 8.1 | 27.9 ± 5.3 | 54.5 ± 4.7 |
| | MedAgents [72] | 79.1 ± 7.4 | 69.7 ± 4.7 | 45.4 ± 8.1 | 39.6 ± 3.0 | 51.6 ± 4.8 |
| | Meta-Prompting [70] | 80.6 ± 1.2 | 73.3 ± 2.3 | 55.3 ± 2.3 | 42.6 ± 4.2 | - |
| Multi-agent (Multi-model) | Reconcile [10] | 81.3 ± 3.0 | **79.7** ± 3.2 | 57.5 ± 3.3 | 31.4 ± 1.2 | - |
| | AutoGen [86] | 60.6 ± 5.0 | 77.3 ± 2.3 | 43.0 ± 8.9 | 37.3 ± 6.1 | - |
| | DyLAN [51] | 64.2 ± 2.3 | 73.6 ± 4.2 | 41.3 ± 1.2 | 34.0 ± 3.5 | - |
| Adaptive | **MDAgents (Ours)** | **88.7** ± 4.0 | 75.0 ± 1.0 | **65.3** ± 3.9 | **56.4** ± 4.5 | 56.2 ± 6.7 |

| Category | Method | Clinical Reasoning and Diagnostic Datasets | | | | |
|---|---|---|---|---|---|---|
| | | DDXPlus | SymCat | JAMA | MedBullets | MIMIC-CXR |
| Single-agent | Zero-shot | 70.3 ± 2.0 | 88.7 ± 2.3 | 62.0 ± 2.0 | 67.0 ± 1.4 | 40.0 ± 5.3 |
| | Few-shot | 69.4 ± 1.0 | 86.7 ± 3.1 | 69.0 ± 4.2 | 72.0 ± 2.8 | 35.3 ± 5.0 |
| | + CoT [83] | 72.7 ± 7.7 | 78.0 ± 2.0 | 66.0 ± 5.7 | 70.0 ± 0.0 | 36.2 ± 5.2 |
| | + CoT-SC [82] | 52.1 ± 6.4 | 83.3 ± 3.1 | 68.0 ± 2.8 | 76.0 ± 2.8 | 51.7 ± 4.0 |
| | ER [67] | 61.3 ± 2.4 | 82.7 ± 2.3 | **71.0** ± 1.4 | 76.0 ± 5.7 | 50.0 ± 0.0 |
| | Medprompt [59] | 59.5 ± 17.7 | 87.3 ± 1.2 | 70.7 ± 4.3 | 71.0 ± 1.4 | 53.4 ± 4.3 |
| Multi-agent (Single-model) | Majority Voting | 67.8 ± 4.9 | 91.9 ± 2.2 | 70.0 ± 5.7 | 70.0 ± 0.0 | 49.5 ± 10.7 |
| | Weighted Voting | 65.9 ± 3.3 | 90.5 ± 2.9 | 66.1 ± 4.1 | 66.0 ± 5.7 | 53.5 ± 2.2 |
| | Borda Count | 67.1 ± 6.7 | 78.0 ± 11.8 | 61.0 ± 5.6 | 66.0 ± 2.8 | 45.3 ± 6.8 |
| | MedAgents [72] | 62.8 ± 5.6 | 90.0 ± 0.0 | 66.0 ± 5.7 | 77.0 ± 1.4 | 43.3 ± 7.0 |
| | Meta-Prompting [70] | 52.6 ± 6.1 | 77.3 ± 2.3 | 64.7 ± 3.1 | 49.3 ± 1.2 | 42.0 ± 4.0 |
| Multi-agent (Multi-model) | Reconcile [10] | 68.4 ± 7.4 | 90.6 ± 2.5 | 60.7 ± 5.7 | 59.5 ± 8.7 | 33.3 ± 3.4 |
| | AutoGen [86] | 67.3 ± 11.8 | 73.3 ± 3.1 | 64.6 ± 1.2 | 55.3 ± 3.1 | 43.3 ± 4.2 |
| | DyLAN [51] | 56.4 ± 2.9 | 75.3 ± 4.6 | 60.1 ± 3.1 | 57.3 ± 6.1 | 38.7 ± 1.2 |
| Adaptive | **MDAgents (Ours)** | **77.9** ± 2.1 | **93.1** ± 1.0 | 70.9 ± 0.3 | **80.8** ± 1.7 | 55.9 ± 9.1 |

\* **CoT**: Chain-of-Thought, **SC**: Self-Consistency, **ER**: Ensemble Refinement
\* 🔵: *text*-only, 🟢: *image+text*, 🟡: *video+text*
\* All experiments were tested with 3 random seeds

# 4 Experiments and Results

In this section, we evaluate our framework and baseline methods across different medical benchmarks in **Solo**, **Group**, and **Aaptive** settings. Our experiments and ablation studies highlight the framework's performance, demonstrating robustness and efficiency by modulating agent numbers and temperatures. Results also show a beneficial convergence of agent opinions in collaborative settings.

## 4.1 Setup

To verify the effectiveness of our framework, we conduct comprehensive experiments with baseline methods on ten datasets including MedQA, PubMedQA, DDXPlus, SymCat, JAMA, MedBullets, Path-VQA, PMC-VQA, MIMIC-CXR and MedVidQA. A detailed explanation and statistics for each dataset are deferred to Appendix A and Figure 8. We use 50 samples per dataset for testing, and the inference time for each complexity was: *low* - 14.7*s*, *moderate* - 95.5*s*, and *high* - 226*s* in average. For all the quantitative experiments in this section, we compare three settings: (1) **Solo**: Using a single LLM agent in the decision-making state. (2) **Group**: Implementing multi-agents to collaborate during the decision-making process. (3) **Adaptive**: Our proposed method MDAgents, adaptively constructs the inference structure from PCP to MDT and ICT. We use 3-shot prompting for low-complexity cases and zero-shot prompting for moderate and high-complexity cases across all settings.

**Medical Question Answering** With MedQA [35], PubMedQA [36], MedBullets [9], and JAMA [9], we focus on question answering through text, involving both literature-based and conceptual medical knowledge questions. Specifically, PubMedQA tasks models to answer questions using abstracts from PubMed, requiring synthesis of biomedical information. MedQA tests the model's ability to understand and respond to multiple-choice questions derived from medical educational materials and examinations. MedBullets provides USMLE Step 2/3 type questions that demand

the application of medical knowledge and clinical reasoning. JAMA Clinical Challenge presents challenging real-world clinical cases with diagnosis or treatment decision-making questions, testing the model's clinical reasoning (Figure 8 in Appendix shows complexity distribution for each dataset)

**Diagnostic Reasoning** DDXPlus [73] and SymCat [2] involve clinical vignettes that require differential diagnosis, closely mimicking the diagnostic process of physicians. These tasks test the model's ability to reason through symptoms and clinical data to suggest possible medical conditions, evaluating the AI's diagnostic reasoning abilities similar to a clinical setting. SymCat [2] uses synthetic patient records constructed from a public disease-symptom data source and is enhanced with additional contextual information through the NLICE method.

**Medical Visual Interpretation** Path-VQA [30], PMC-VQA [95], MedVidQA [29], and MIMIC-CXR [3] challenge models to interpret medical images and videos, requiring integration of visual data with clinical knowledge. PathVQA focuses on answering questions based on pathology images, testing AI's capability to interpret complex visual information from medical images. PMC-VQA evaluates AI's proficiency in deriving answers from both text and images found in scientific publications. MedVidQA extends to video-based content, where AI models need to process information from medical procedure videos. MIMIC-CXR-VQA specifically targets chest radiographs, utilizing a diverse and large-scale dataset designed for visual question-answering tasks in the medical domain.

**Baseline Methods**

- **Solo:** The baseline methods considered for the Solo setting include the following: Zero-shot [41] directly incorporates a prompt to facilitate inference, while Few-shot [8] involves a small number of examples. Few-shot CoT [83] integrates rationales before deducing the answer. Few-shot CoT-SC [82] builds upon Few-shot CoT by sampling multiple chains to yield the majority answer. Ensemble Refinement (ER) [67] is a prompting strategy that conditions model responses on multiple reasoning paths to bolster the reasoning capabilities of LLMs. Medprompt [59] is a composition of several prompting strategies that enhances the performance of LLMs and achieves state-of-the-art results on multiple benchmark datasets, including medical and non-medical domains.

- **Group:** We tested five group decision-making methods: Voting [82], MedAgents [72], Reconcile [10], AutoGen [86], and DyLAN [51]. Autogen was based on four agents, with one User, one Clinician, one Medical Expert, and one Moderator, with one response per agent [86]. DyLAN setup followed the base implementations of four agents with no specific roles and four maximum rounds of interaction [51]. While the methods support multiple models, GPT-4 was used for all agents.

### 4.2 Results

In Table 2, we report the classification accuracy on MedQA, PubMedQA, DDXPlus, SymCat, JAMA, MedBullets, Path-VQA, PMC-VQA and MedVidQA dataset. We compare our method (Adaptive) with several baselines in both Solo and Group settings.

**Adaptive method outperforms Solo and Group settings.** As depicted in Figure 4 and Table 2, MDAgents significantly outperforms (p < 0.05) both Solo and Group setting methods, showing best performance in **7 out of 10** medical benchmarks tested. This reveals the effectiveness of adaptive strategies integrated within our system, particularly when navigating through the text-only (e.g., DDXPlus where it outperformed the best performance of single-agent by 5.2% and multi-agent by 9.5%) and text-image datasets (e.g., Path-VQA, PMC-VQA and MIMIC-CXR). Our approach not only comprehends textual information with high precision but also adeptly synthesizes visual data, a pivotal capability in medical diagnostic evaluations.

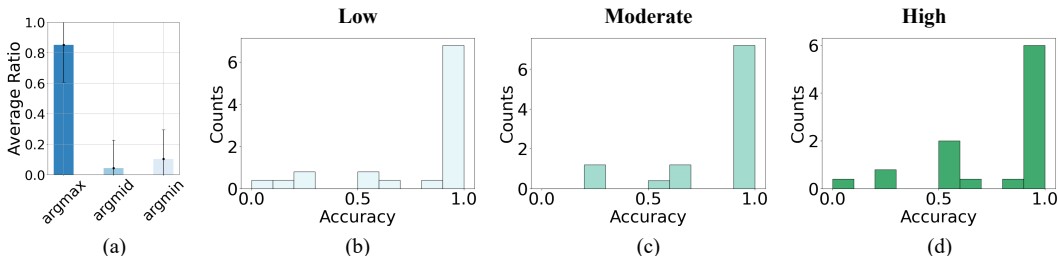

Figure 3: Experiment with the MedQA dataset (*N*=25 randomly sampled questions). (a) LLM's capability to classify complexity. (b-d) Evaluating 25 medical problems by solving each one 10 times at various complexity levels. The x-axis represents the accuracy achieved for each problem, while the y-axis shows the number of problems that reached that level of accuracy.

**Why Do Adaptive Decision-making Framework Work Well?**    It is important to accurately assign difficulty levels to medical questions. For instance, if a medical question is obviously easy, utilizing a team of specialists (such as an IDT) might be excessive and potentially lead to overly pessimistic approaches. Conversely, if a difficult medical question is only tackled by a PCP, the problem might not be adequately addressed. The core issue here is the LLM's capability to classify the difficulty of medical questions appropriately. If an LLM inaccurately classifies the difficulty level, the chosen medical solution may not be suitable, potentially leading to the wrong decision making. Therefore, understanding what constitutes an appropriate difficulty level is essential.

We hypothesize that an LLM, functioning as a classifier, will select the optimal complexity level for each MDM problem. This hypothesis is supported by Figure 3, which illustrates that the model appropriately matches the complexity levels; low, moderate, and high of the given problem. To determine this, we assessed the accuracy of solutions across various difficulty levels. Specifically, we evaluated 25 medical problems by repeating each problem for 10 times at each difficulty level. By measuring the success rate, we aimed to identify the difficulty level that yielded the highest accuracy. This approach ensures that the LLM's complexity classification aligns with the most effective and accurate medical solutions, thereby optimizing the application of medical expertise to each question.

Formally, for any given problem $P$, we denote the probability that the correct answer can be solved at a specific complexity level as $p_{\text{complexity-level}}(P)$, where complexity-level $\in \{\text{low}, \text{moderate}, \text{high}\}$. $\arg\max(P) \in \{\text{low}, \text{moderate}, \text{high}\}$ refers to the complexity level that has the highest probability among $p_{\text{low}}(P)$, $p_{\text{moderate}}(P)$, and $p_{\text{high}}(P)$. Similarly, $\arg\min(P)$ is the complexity level with the lowest probability, and $\arg\text{mid}(P)$ is the one with the middle probability. We denote $a$, $b$, and $c$ as the probabilities that the LLM selects the complexity levels corresponding to $\arg\max$, $\arg\text{mid}$, and $\arg\min$, respectively. Thus, the accuracy of our system for problem $P$ can be described by $a \cdot p_{\arg\max}(P) + b \cdot p_{\arg\text{mid}}(P) + c \cdot p_{\arg\min}(P)$, and the overall accuracy is given by $\mathbb{E}_P \left[ a \cdot p_{\arg\max}(P) + b \cdot p_{\arg\text{mid}}(P) + c \cdot p_{\arg\min}(P) \right]$. The estimated values of $a, b, c$ are $a = 0.81 \pm 0.29$, $b = 0.11 \pm 0.28$, and $c = 0.08 \pm 0.16$, which indicates that LLM can provide an optimal complexity level with probability at least $80\%$.

These findings suggest that a classifier LLM can implicitly simulate various complexity levels and optimally adapt to the complexity required for each medical problem, as shown in Figure 3. This ability to adjust complexity dynamically proves to be crucial for applying LLMs effectively in MDM contexts as shown by the competitiveness of our Adaptive approach.

**Solo vs. Group Setting in MDM.**    The experimental results reveal distinct performance patterns between Solo and Group settings across various medical benchmarks. In simpler datasets like MedQA, solo methods, leveraging Few-shot CoT and CoT-SC, achieved up to 83.9% accuracy compared to the group's best of 81.3%. Conversely, for more complex datasets like Sym-Cat, group settings perform better, with SymCat showing 91.9% accuracy in the group settings versus 88.7% in solo settings. Notably, group settings (e.g. Weighted Voting, Reconcile) performed better in multi-modal datasets such as Path-VQA (*image + text*), MedVidQA (*video + text*), and MIMIC-CXR (*image + text*), high-

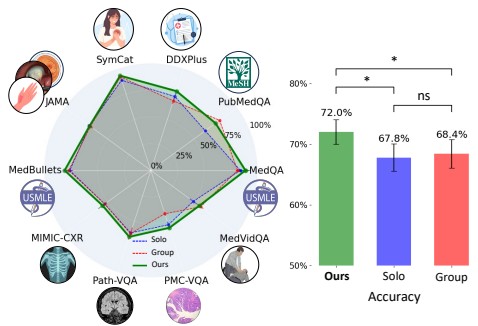

Figure 4: Our method outperforms Solo and Group settings across different medical benchmarks.

lighting the advantage of collaborative process in complex tasks. This result aligns with findings from [4], which showed that pooled diagnoses from groups of physicians significantly outperformed those of individual physicians, with accuracy increasing as group size increased. Overall, solo settings outperformed group settings in four benchmarks, while group settings outperformed solo in six benchmarks. These results reveals that while solo methods excel in straightforward tasks, group settings provide better performance in complex, multi-faceted tasks requiring diverse expertise.

### 4.3 Ablation Studies

**Impact of Complexity Selection.**    We evaluate the importance of the complexity assessment and adaptive process through ablation studies (Figure 5). Our adaptive method significantly outperforms static complexity assignments across different modality benchmarks. For *text*-only queries, the Adaptive method achieves an accuracy of 81.2%, significantly higher than *low* (64.2%), *moderate* (71.6%), and *high* (65.8%) settings. Interestingly, 64% of the text-only queries were classified as

*high* complexity, indicating that many text-based queries required in-depth analysis with different expertise. In the *image + text* modality, the Adaptive method classified 55% of the queries as *low* complexity, suggesting that the visual information often provides clear and straightforward cues that simplify the decision-making process. Finally, for *video + text* queries, 87% of these queries were classified as *low* complexity, reflecting that the dynamic visual data in conjunction with text can often be straightforwardly interpreted. However, further evaluation on more challenging video medical datasets is needed, as MedVidQA contains relatively less complex medical knowledge.

**Impact of Moderator's Review and RAG** Table 3 examines the impact of incorporating external medical knowledge and moderator reviews into the MDAgents framework on accuracy. MedRAG [89] is a systematic toolkit for Retrieval-Augmented Generation (RAG) that leverages various corpora; biomedical, clinical and general medicine, to provide

| Method | Avg. Accuracy (%) |
|---|---|
| MDAgents (Ours) | 71.8 |
| + MedRAG | 75.2 (↑ **4.7 %**) |
| + Moderator's Review | 77.6 (↑ **8.1 %**) |
| + Moderator's Review & MedRAG | 80.3 (↑ **11.8 %**) |

Table 3: Ablations for the impact of moderator's review and MedRAG. The Accuracy were averaged accuracy across all datasets.

comprehensive knowledge. The baseline accuracy of MDAgents is 71.8%. Integrating MedRAG increases accuracy to 75.2% (up 4.7%), while the moderator's review alone raises it to 77.6% (up 8.1%). The combined use of both methods achieves the highest accuracy at 80.3% (up 11.8%).

The results indicate that MedRAG and moderator review both enhance performance, with their combined effect being synergistic. This highlights that leveraging recent external knowledge and structured feedback mechanisms is crucial for refining and converging on accurate medical decisions. This improvement underscores the importance of a hybrid strategy, aligning with real-world practices of continuous learning and expert consultation to optimize performance in medical applications.

## 4.4 Impact of Number of Agents in Group Setting.

Our experiment with varying the number of agents in a collaborative Group setting (Appendix Figure 6 (a-b)) shows that a higher number of agents does not lead to better performance. Rather, our Adaptive method achieves optimal performance with fewer agents (*N*=3, peak accuracy of 83.5%) by intelligently calibrating the number of collaborating agents. This not only indicates efficiency in decision-making but also computational and economic benefits, considering the reduced number of API calls needed, especially when contrasted with the Solo and Group settings.

With regards to computational efficiency, the Solo setting (5-shot CoT-SC) resulted in a 6.0 and Group setting (MedAgents with *N*=5) resulted in a 20.3 API calls, suggesting a high computational cost without a corresponding increase in accuracy. On the other hand, our Adaptive method exhibits a more economical use of resources, demonstrated by fewer API calls (9.3 with *N*=3) while maintaining high accuracy, a critical factor in deploying scalable and cost-effective medical AI solutions.

## 4.5 Robustness of MDAgents with different parameters.

Our Adaptive approach shows resilience to changes in temperature (Appendix Figure 6 (c), low (*T*=0.3) and high (*T*=1.2)) with performance improving under higher temperatures. This suggests that our model can utilize the creative and diverse outputs generated at higher temperatures to enhance decision-making, a property that is not as pronounced in the Solo and Group conditions. This robustness is particularly valuable in real-world medical domains with high uncertainty and ambiguity in datasets [15]. Additionally, studies have shown that creative diagnostic approaches

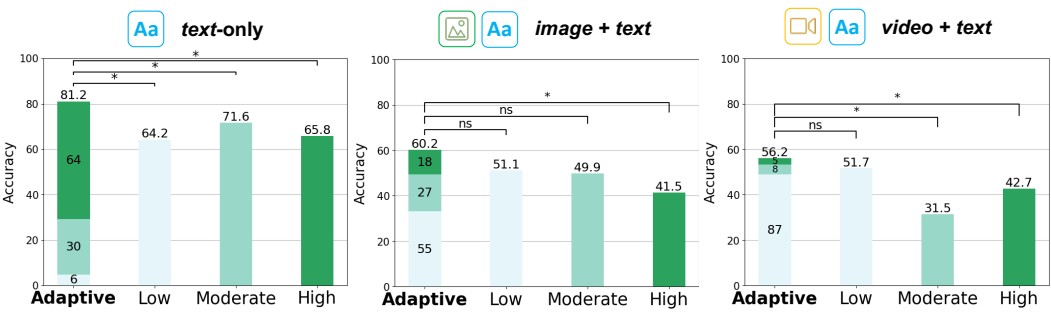

Figure 5: Impact of complexity selection of the query. Accuracy of each ablation on *text*-only (left), *text+image* (center) and *text+video* (right) benchmarks are reported.

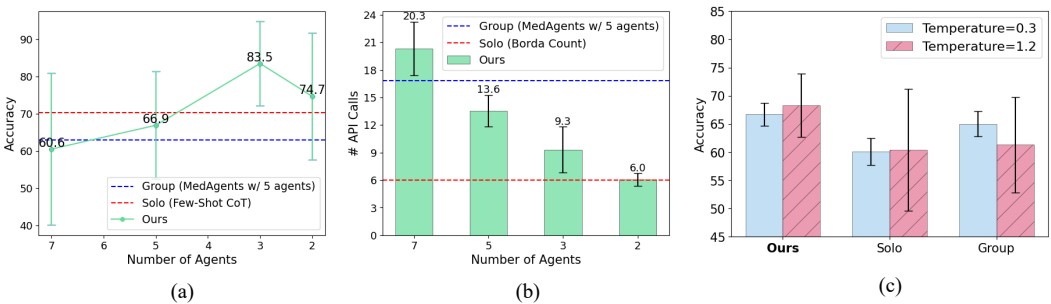

(a)                                     (b)                                     (c)

Figure 6: Impact of the number of agents on (a) Accuracy, (b) Number of API Calls on medical benchmarks with GPT-4 (V) and (c) Performance of three different settings under low ($T$=0.3) and high ($T$=1.2) temperatures on medical benchmarks. Our Adaptive setting shows better robustness to different temperatures and even takes advantage of higher temperatures.

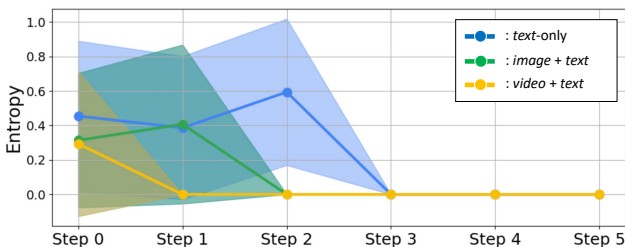

Figure 7: An illustration of consensus entropy in group collaboration process of MDAgents (w/ Gemini-Pro (Vision), $N$=30 for each dataset) on medical benchmarks with different modality inputs.

can mitigate cognitive biases and improve diagnostic accuracy [66], while fostering flexibility and adaptability in decision-making [19]. These insights support the enhanced performance observed under higher temperatures in our framework. However, the future work should explore a wider range of temperatures to fully understand the robustness and adaptability of our approach.

## 4.6 Convergence Trends in Consensus Dynamics

There is clear trend towards consensus among MDAgents cross various data modalities (Figure 7). The *text+video* modality demonstrates a rapid convergence, reflecting the agents' efficient processing of combined textual and visual cues. On the other hand, the *text+image* and *text*-only modalities display a more gradual decline in entropy, indicating a progressive narrowing of interpretative diversity among the agents. Despite the differing rates and initial conditions, all modalities exhibit convergence of agent opinions over time. This uniformity in reaching consensus highlights the MDAgents' capability to integrate and reconcile information. Please refer to Appendix B for a detailed explanation of the entropy calculation.

## 5 Conclusion

This paper introduces **MDAgents**, a framework designed to enhance the utility of LLMs in complex medical decision-making by dynamically structuring effective collaboration models. To reflect the nuanced consultation aspects in clinical settings, MDAgents adaptively assigns LLMs either to roles independently or within groups, depending on the task's complexity. This emulation of real-world medical decision processes has been comprehensively evaluated, with MDAgents outperforming previous solo and group methods in **7 out of 10** medical benchmarks. The case study illustrates the practical efficacy and collaborative dynamics of our proposed framework, providing insights into how differing expert opinions are synthesized to reach a more accurate diagnosis. This is evidenced by our agents' ability to converge on the correct diagnosis despite initially divergent perspectives. Ablation studies further elucidate the individual contributions of agents and strategies within the system, revealing the critical components and interactions that drive the framework's success. By harnessing the strength of multi-modal reasoning and fostering a collaborative process among LLM agents, our framework opens up new possibilities for enhancing LLM-assisted medical diagnosis systems, pushing the boundaries of automated clinical reasoning.

## Acknowledgments and Disclosure of Funding

We thank Yoon Kim at MIT, Vivek Natarajan at Google, WonJin Yoon and Tim Miller at Harvard Medical School, Seonghwan Bae at Sacheon Public Health Center, Hui Dong Lim at Seoul National University Hospital for their revisions, feedback, and support. C.P. acknowledges support from the Takeda Fellowship, the Korea Foundation for Advanced Studies, and the Siebel Scholarship.

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

## Limitations and Future Works

Despite the successes of our framework in showing promising performance in medical decision-making tasks, we recognize several limitations that open pathways for future research.

**Medical Focused Foundation Models.** An essential enhancement would be to incorporate the foundation models and systems specifically trained on medical data, like Med-Gemini [63], AMIE [77], and Med-PaLM 2 [67]. These models excel in generating professional medical terminologies, which can facilitate more effective and accurate communication between multiple agents involved in the decision-making process. By leveraging these specialized models, the agents can interact using a shared, precise medical vocabulary, enhancing the system's overall performance and reliability. This approach not only ensures more medically accurate content generation but also supports better collaboration and understanding among the agents, which is essential for complex medical decision-making tasks.

**Patient-Centered Diagnosis.** A primary limitation lies in the fact that our current framework operates within the confines of multiple-choice question answering and does not account for the interactive, patient-centered nature of real-world diagnostics. Effective diagnosis often relies on continuous exchanges that include the patient's narrative, the physician's expertise, and input from caregivers. To bridge this gap, future iterations of our framework will aim to incorporate a more interactive system that not only assists physicians but also directly engages with both patients and caregivers in a multi-stakeholder approach. Moreover, by incorporating regret-aware [91] decision-making, the system can learn to minimize diagnostic regret over time, refining its responses based on the outcomes of prior interactions. This regret-aware framework will help guide the LLM to seek additional information when uncertainties arise, thereby supporting more informed decisions across complex, multi-stakeholder scenarios. Embedding these real-world interactions within the feedback loop will enable the system to provide more nuanced and patient-centric support, enhancing the quality and personalization of medical decision-making across all involved parties.

**Potential Risks and Mitigations.** While our framework shows promise, potential risks include medical hallucinations and the generation of inaccurate or misleading information. To address these risks, integrating self-correction mechanisms, such as those proposed by [42], could enable the model to autonomously identify and rectify its own errors via reinforcement learning-based self-correction. Additionally, implementing rule-based reward structures, as suggested in [56], would allow the model to adhere to specific safety and accuracy guidelines during training. These methods can support a safer, more reliable diagnostic support tool by introducing corrective feedback loops and standardized behavior guidelines. Furthermore, integrating confidence scores and uncertainty estimates with the model's recommendations could enhance the decision-making process by enabling end-users to weigh various diagnostic options, ultimately increasing the system's trustworthiness and safety.

## A  Dataset Information

We evaluate multi-agent collaboration frameworks across seven common medical question-answering datasets, which vary in question complexity. Generally, questions are deemed more complex if they involve multiple modalities or entail a lengthy, detailed diagnostic task. Below, we detail each dataset and provide a sample entry:

1. **MedQA.** The MedQA dataset consists of professional medical board exams from the US, Mainland China, and Taiwan [35]. Our study focuses on the English test set, comprising 1,273 questions sourced from the United States Medical Licensing Examination (USMLE). These questions are formatted as multiple-choice text queries with five options. Due to their textual nature and brevity, we categorize these questions as low.
   Sample Question: *"A 47-year-old female undergoes a thyroidectomy for treatment of Graves' disease. Post-operatively, she reports a hoarse voice and difficulty speaking. You suspect that this is likely a complication of her recent surgery. What is the embryologic origin of the damaged nerve that is most likely causing this patient's hoarseness?"*
   Options: *A: 1st pharyngeal arch, B: 2nd pharyngeal arch, C: 3rd pharyngeal arch, D: 4th pharyngeal arch, E: 6th pharyngeal arch*

Table 4: Summary of the Datasets. (T): Text, (I): Image, (V): Video. In Appendix A, we provide detailed sample information for each benchmark.

| Dataset | Modality | Format | Choice | Testing Size | Domain |
|---|---|---|---|---|---|
| MedQA | T | Question + Answer | A/B/C/D | 1273 | US Medical Licensing Examination |
| PubMedQA | T | Question + Context + Answer | Yes/No/Maybe | 500 | PubMed paper abstracts |
| DDxPlus | T | Question + Answer | A/B/C/D/ · · · | 134 K | Pathologies, Symptoms and Antecedents from Patients |
| SymCat | T | Question + Answer | A/B/C/D | 369 K | Disease-symptom records from public datasets |
| JAMA | T | Case + Question + Answer | A/B/C/D | 1524 | Challenging real-world Clinical Cases from diverse Medical Domains |
| MedBullets | T | Case + Question + Answer | A/B/C/D | 308 | Online Platform Resources for Medical Study |
| MIMIC-CXR | T I | Question + Answer | Closed Answer | 1531 | Chest X-ray images and Free-text Reports. |
| PMC-VQA | T I | Question + Answer | A/B/C/D | 50 K | VQA pairs across Images, spanning diverse Modalities and Diseases |
| Path-VQA | T I | Question + Answer | Yes/No | 3391 | Open-ended Questions from Pathology Images |
| MedVidQA | T V | Question + Answer | A/B/C/D | 155 | First Aids, Medical Emergency, and Medical Education Questions |

2. **PubMedQA.** PubMedQA is a QA dataset based on biomedical research [36]. It requires yes/no/maybe answers to questions grounded in PubMed abstract. The dataset comprises entries each containing a question, a relevant abstract minus the conclusion, and a ground truth label. We used 50 samples for testing. Given its binary choice format, we consider the complexity of this dataset to be low.
Sample Question: *"Can predilatation in transcatheter aortic valve implantation be omitted?"*
Context: *"The use of a balloon expandable stent valve includes balloon predilatation of the aortic stenosis before valve deployment. The aim of the study was to see whether or not balloon predilatation is necessary in transcatheter aortic valve replacement (TAVI). Sixty consecutive TAVI patients were randomized to the standard procedure or to a protocol where balloon predilatation was omitted. There were no significant differences between the groups regarding early hemodynamic results or complication rates."*

3. **DDXPlus.** DDXPlus is a medical diagnosis dataset using synthetic patient information and symptoms [73]. Each instance represents a patient, with attributes including age, sex, initial evidences, evidence, multiple options of possible pathologies, and a ground truth diagnosis. Due to its text-only and multiple-choice nature, we consider the complexity to be low.
Sample Patient Information: *Age: 96, Sex: F*
Evidences: *['e66', 'insp_siffla', 'j45', 'posttus_emesis', 'trav1_@_N', 'vaccination']*
Initial evidence: *'posttus_emesis'*
Options: *(A) Bronchite (B) Coqueluche*

4. **PMC-VQA.** PMC-VQA is a large-scale medical visual question-answering dataset that contains 227K Visual Question Answering (VQA) pairs of 149K images [95]. It is structured as a multiple-choice QA task with one image input accompanying each question. Since the query requires a model to consider both text and image inputs, while maintaining medical expertise, we consider the complexity to be moderate.
Sample Question: *What is the appearance of the hyperintense foci in the basal ganglia on T1-weighted MRI image?*
Image: *PMC8415802_FIG1.jpg*
Options: *A: Hypodense, B: Hyperdense, C: Isointense, D: Hypointense*

5. **Path-VQA.** PathVQA is a VQA dataset specifically on pathology images [30]. Different from PMC-VQA which consists of multiple choice questions, Path-VQA includes open-ended questions and binary "yes/no" questions. For the purpose of maintaining a standardized accuracy evaluation, we use only the yes/no questions. Similar to PMC-VQA, we consider the complexity to be moderate.
Sample Question: *Was a gravid uterus removed for postpartum bleeding?*
Image: *test_0273.jpg*

6. **MedVidQA.** MedVidQA dataset consists of 3,010 health-related questions with visual answers from validated video sources (e.g. medical school, health institutions, etc). We

enhanced the dataset by using GPT-4 to generate multiple-choice answers, including one correct 'golden answer' and several false options, expanding its use for training and evaluating automated medical question-answering systems.
Sample Question: *How to perform corner stretches to treat neck pain?*
Video: *h5MvX50zTLM.mp4*
Options: *A: By bending your knees and touching your toes, B: By performing jumping jacks, C: By leaning into a corner with your elbows up at shoulder level, D: By doing push-ups*

7. **SymCat.** SymCat is a synthetic dataset which includes 5 million symptom-condition samples, covering 801 distinct conditions each with 376 potential symptoms dataset [2].
*Sample Patient Information: Age: 42, Gender: F, Race: White, Ethnicity: Nonhispanic*
*Sample Patient Symptoms: Pain:cramping:Abdomen:Lower abdomen::::Worsening after meals::32; Altered stool:Lumpy:::::::Often:44; Flatulence:::::::::30; Bloating:::::::::41*
*Options: "A": "Asthma", "B": "IBS (Constipation type)", "C": "Viral meningitis (Varicella zoster virus)", "D": "Bacterial (Gastro)enteritis (Yersinia infection most likely)"*

8. **JAMA.** JAMA includes 1524 clinical cases collected from the JAMA Network Clinical Challenge archive, which are summaries of actual challenging clinical cases. Each sample is framed as a question, with a long case description and four options [9].
*Sample Case: A 62-year-old woman undergoing peritoneal dialysis (PD) for kidney failure due to IgA nephropathy presented to the PD clinic with a 1-day history of severe abdominal pain and cloudy PD fluid. Seven days prior, she inadvertently broke aseptic technique when tightening a leaking connection of her PD catheter tubing. On presentation, she was afebrile and had normal vital signs. Physical examination revealed diffuse abdominal tenderness. Cloudy fluid that was drained from her PD catheter was sent for laboratory analysis (Table 1).Await peritoneal dialysis fluid culture results before starting intraperitoneal antibiotics.*
*Sample Question: What Would You Do Next?*
*"A": "Administer empirical broad-spectrum intraperitoneal antibiotics", "B": "Administer empirical broad-spectrum intravenous antibiotics", "C": "Await peritoneal dialysis fluid culture results before starting intraperitoneal antibiotics", "D": "Send blood cultures"*

9. **Medbullets.** Medbullets comprises 308 USMLE Step 2/3 style questions collected from open-access tweets on X (formerly Twitter) since April 2022. The difficulty is comparable to that of Step 2/3 exams, which emulate common clinical scenarios [9].
*Sample Question: A 2-week-old boy is evaluated by his pediatrician for abnormal feet. The patient was born at 39 weeks via vaginal delivery to a G1P1 29-year-old woman. The patient has been breastfeeding and producing 5 stools/day. He is otherwise healthy. His temperature is 99.5ŏb0F (37.5ŏb0C), blood pressure is 60/38 mmHg, pulse is 150/min, respirations are 24/min, and oxygen saturation is 98% on room air. A cardiopulmonary exam is notable for a benign flow murmur. A musculoskeletal exam reveals the findings shown in Figure A. Which of the following is the most appropriate next step in management? Options: "A": "Botulinum toxin injections", "B": "Reassurance and reassessment in 1 month", "C": "Serial casting", "D": "Surgical pinning",*

10. **MIMIC-CXR-VQA.** MIMIC-CXR is a large-scale visual question-answering dataset of 377,110 chest radiographs. It was obtained from 227,827 imaging studies sourced from the BIDMC between 2011-2016. It includes patient identifiers which can be linked to MIMIC-IV [3].
*Sample Question: Are there any abnormalities in the upper mediastinum?*
*Image: p11/p11218589/s59138139/7a1e4762-c176bd78-6281fe5c-6b0c9734-e9a4c8f1.jpg*

## B Entropy Calculation for Consensus Dynamics

The entropy $H$ serves as an indicator of consensus progression among the agents. It is quantified as:

$$H = -\sum_{i=1}^{M} p(x_i) \log_2 p(x_i) \tag{B.1}$$

where $M$ is the total number of unique answers, $x_i$ represents a unique answer, and $p(x_i)$ is the probability of occurring among all answers. This calculation helps to measure the degree of agreement among the agents over time, with lower entropy indicating higher consensus. The trends in entropy

across different data modalities provide insights into how quickly and effectively MDAgents can reach a unified decision.

## C   Prompt Templates

### C.1   A single agent setting

> **Few-shot multiple choice questions**
>
> `{{instruction}}`
> The following are multiple choice questions (with answers) about medical knowledge.
> `{{few_shot_examples}}`
> `{{context}}` **Question:** `{{question}}` `{{answer_choices}}` **Answer:**(

> **Chain-of-Thought multiple choice questions**
>
> `{{instruction}}`
> The following are multiple choice questions (with answers) about medical knowledge.
> `{{few_shot_examples w/ CoT Solutions}}`
> `{{context}}` **Question:** `{{question}}` `{{answer_choices}}` **Answer:**(

> **Ensemble Refinement multiple choice questions**
>
> `{{instruction}}`
> The following are multiple choice questions (with answers) about medical knowledge.
> `{{few_shot_examples w/ CoT Solutions}}`
> `{{context}}` **Question:** `{{question}}` `{{answer_choices}}`
> `{{reasoning_paths}}` **Answer:**(

> **Medprompt multiple choice questions**
>
> `{{instruction}}`
> The following are multiple choice questions (with answers) about medical knowledge.
> `{{few_shot_examples w/ CoT Solutions from similarity calculation}}`
> for $N$ times do
>   `{{context}}` **Question:** `{{question}}` `{{shuffled_answer_choices}}`
>   **Answer:**(

### C.2   Multi-agent setting

> **Complexity check prompt**
>
> You are a medical expert who conducts initial assessment and your job is to decide the difficulty/complexity of the medical query.
> Now, given the medical query as below, you need to decide the difficulty/complexity of it:
> `{{question}}`
> Please indicate the difficulty/complexity of the medical query among below options:
> 1) low: a PCP or general physician can answer this simple medical knowledge checking question without relying heavily on consulting other specialists.
> 2) moderate: a PCP or general physician can answer this question in consultation with other specialist in a team.
> 3) high: Team of multi-departmental specialists can answer to the question which requires specialists consulting to another department (requires a lot of team effort to treat the case).
> **Answer:**(

> **Recruiter prompt**
>
> You are an experienced medical expert who recruits a group of experts with diverse identity and ask them to discuss and solve the given medical query.
> Now, given the medical query as below, you need to decide the difficulty/complexity of it:
> `{{question}}`
> You can recruit up to *N* experts in different medical expertise. Considering the medical question and the options for the answer, what kind of experts will you recruit to better make an accurate answer?
> Also, you need to specify the communication structure between experts (e.g., Pulmonologist == Neonatologist == Medical Geneticist == Pediatrician > Cardiologist)
>
> For example, if you want to recruit five experts, you answer can be like:
> `{{examplers}}`
> Please answer in above format, and do not include your reason.
> **Answer:**(

# D  Additional Results

This section presents additional experimental results and analyses that provide further insights into the performance and characteristics of our MDAgents framework.

> **Agent initialization prompt**
>
> You are a `{{role}}` who `{{description}}`. Your job is to collaborate with other medical experts in a team.

> **Agent interaction prompt**
>
> Given the opinions from other medical agents in your team, please indicate whether you want to talk to any expert (yes/no). If not, provide your opinion. `{{opinions}}`
>
> Next, indicate the agent you want to talk to: `{{agent_list}}`
>
> Remind your medical expertise and leave your opinion to an expert you chose. Deliver your opinion once you are confident enough and in a way to convince other expert with a short reason.

## D.1 Accuracy on entire MedQA 5-options Dataset

To provide a comprehensive evaluation of our approach, we conducted experiments on the entire MedQA 5-options dataset using GPT-4o mini. This expands upon the subsampled experiments presented in the main experiments in Table 2. Below Table shows the accuracy results for various methods.

Table 5: Accuracy (%) on entire MedQA 5-options dataset with GPT-4o mini

| Category | Method | Accuracy (%) |
|---|---|---|
| Single-agent | Zero-shot | 71.5 |
| | 3-shot | 72.3 |
| | + CoT | 76.6 |
| | + CoT-SC | 77.2 |
| Multi-agent (Multi-model) | Majority Voting | 76.3 |
| | Weighted Voting | 79.1 |
| | Borda Count | 76.1 |
| Multi-agent (Single-model) | Reconcile | 80.2 |
| Adaptive | **MDAgents (Ours)** | **83.6** |

These results demonstrate that our MDAgents approach outperforms both single-agent and other multi-agent methods across the full dataset, achieving an accuracy of 83.6%. This underscores the effectiveness of our framework in handling diverse medical questions at scale.

## D.2 Estimated Costs for Full Test Set Experiments

To provide transparency and aid in reproducibility, we estimated the costs associated with running experiments on the entire test sets using GPT-4 (Vision). Below Table presents these cost estimates in USD for various datasets and methods.

Table 6: Estimated costs for experimenting with entire test sets with GPT-4 (Vision) (in USD)

| Method | MedQA | PubMedQA | Path-VQA | PMC-VQA | DDXPlus | SymCat | JAMA | MedBullets | MIMIC-CXR | Total Cost |
|---|---|---|---|---|---|---|---|---|---|---|
| CoT | 55.24 | 13.16 | 3,028.54 | 27,134.00 | 16,461.90 | 10,593.99 | 134.55 | 61.23 | 1,388.70 | 58,871.29 |
| **Ours** | 172.43 | 41.36 | 9,369.45 | 82,194.34 | 44,814.97 | 31,176.05 | 367.13 | 161.70 | 4,406.90 | 172,704.33 |

While our approach incurs higher costs due to its multi-agent nature, the significant performance improvements justify this increased computational expense for critical medical decision-making tasks.

### D.3 Impact of Knowledge Enhancement with RAG

| Method | Accuracy (%) |
|---|---|
| MDAgents (baseline) | 71.8 |
| + MedRAG | 75.2 |
| + Medical Knowledge Initialization | 76.0 |
| + Moderator's Review | 77.6 |
| + Moderator's Review & MedRAG | **80.3** |

Table 7: Impact of knowledge enhancement on MDAgents performance

We investigated whether simply assigning roles to agents is sufficient for expert-like performance, and explored the impact of equipping agents with different knowledge using Retrieval-Augmented Generation (RAG). Table 7 presents the results of these experiments.

These results indicate that while role assignment provides a foundation, augmenting agents with specific knowledge (using MedRAG) and structured reviews (Moderator's Review) significantly enhances their ability to simulate domain expertise. The combination of Moderator's Review and MedRAG yielded the best performance, highlighting the synergy between structured collaboration and domain-specific knowledge retrieval.

### D.4 Complexity Assignment and Collaborative Settings

To address the impact of complexity assignment on accuracy and API costs, we conducted additional experiments focusing on high-complexity cases, particularly in image+text scenarios. Table 8 shows the results for various collaborative settings.

| Collaboration Setting | Accuracy (%) |
|---|---|
| Sequential & No Discussion | 39.0 |
| Sequential & Discussion | 45.0 |
| Parallel & No Discussion | 56.0 |
| Parallel & Discussion | **59.0** |

Table 8: Impact of collaboration settings on high-complexity image+text tasks

These results underscore the importance of multi-turn discussions, particularly in complex cases. The parallel collaboration with discussion yielded the highest accuracy (59.0%), suggesting that enabling agents to work simultaneously and engage in dialogue is crucial for handling intricate medical queries. The significant performance gap between discussion and no-discussion scenarios (45.0% vs. 39.0% for sequential, and 59.0% vs. 56.0% for parallel) highlights the value of interactive deliberation in medical decision-making processes.

Table 9: Accuracy (%) on Medical benchmarks with **Solo** (🤖) setting. **Bold** represents the best and Underlined represents the second best performance for each benchmark and model.

| Category | Method | Medical Knowledge Retrieval Datasets | | | | |
|---|---|---|---|---|---|---|
| | | MedQA (T) | PubMedQA (T) | Path-VQA (I)(T) | PMC-VQA (I)(T) | MedVidQA (Y)(T) |
| GPT-3.5 | Zero-shot | 48.5 ± 3.3 | 56.8 ± 12.0 | - | - | - |
| | Few-shot | 47.8 ± 16.4 | 59.0 ± 1.0 | - | - | - |
| | + CoT | 54.2 ± 8.9 | 49.7 ± 11.9 | - | - | - |
| | + CoT-SC | 60.5 ± 3.2 | 49.4 ± 13.5 | - | - | - |
| | ER | 60.2 ± 4.0 | 52.9 ± 15.9 | - | - | - |
| | Medprompt | 60.1 ± 10.8 | 59.8 ± 8.5 | - | - | - |
| GPT-4(V) | Zero-shot | 75.0 ± 1.3 | 61.5 ± 2.2 | 57.9 ± 1.6 | 49.0 ± 3.7 | - |
| | Few-shot | 72.9 ± 11.4 | 63.1 ± 11.7 | 57.5 ± 4.5 | 52.2 ± 2.0 | - |
| | + CoT [83] | 82.5 ± 4.9 | 57.6 ± 9.2 | 58.6 ± 3.1 | 51.3 ± 1.5 | - |
| | + CoT-SC [82] | **83.9** ± 2.7 | 58.7 ± 5.0 | 61.2 ± 2.1 | 50.5 ± 5.2 | - |
| | ER [67] | 81.9 ± 2.1 | 56.0 ± 7.0 | 61.4 ± 4.1 | 52.7 ± 2.9 | - |
| | Medprompt [59] | 82.4 ± 5.1 | 51.8 ± 4.6 | 59.2 ± 5.7 | **53.4** ± 7.9 | - |
| Gemini-Pro(Vision) | Zero-shot | 42.0 ± 10.4 | **65.2** ± 14.5 | 45.9 ± 2.8 | 44.8 ± 2.0 | 37.9 ± 8.4 |
| | Few-shot | 34.0 ± 7.2 | 55.0 ± 0.0 | 64.5 ± 2.3 | 48.2 ± 1.0 | 47.1 ± 8.6 |
| | + CoT | 50.0 ± 6.0 | 60.2 ± 9.0 | **66.4** ± 11.7 | 47.1 ± 4.2 | 48.6 ± 5.5 |
| | + CoT-SC | 52.7 ± 4.6 | 55.8 ± 8.9 | 63.6 ± 6.0 | 46.3 ± 2.8 | **49.2** ± 8.2 |
| | ER | 52.0 ± 7.2 | 58.4 ± 14.2 | 57.6 ± 8.4 | 38.4 ± 2.0 | 48.5 ± 4.1 |
| | Medprompt | 45.3 ± 3.1 | 50.6 ± 5.4 | 55.0 ± 2.0 | 41.8 ± 3.0 | 44.5 ± 2.0 |

| Category | Method | Clinical Reasoning and Diagnostic Datasets | | | | |
|---|---|---|---|---|---|---|
| | | DDXPlus (T) | SymCat (T) | JAMA (T) | MedBullets (T) | MIMIC-CXR (I)(T) |
| GPT-3.5 | Zero-shot | 56.2 ± 4.1 | 84.0 ± 0.0 | 36.0 ± 3.3 | 56.0 ± 2.8 | - |
| | Few-shot | 48.9 ± 8.5 | 86.0 ± 2.8 | 38.0 ± 4.2 | 59.0 ± 1.4 | - |
| | + CoT | 52.8 ± 5.4 | 82.0 ± 0.0 | 34.0 ± 2.4 | 56.0 ± 5.7 | - |
| | + CoT-SC | 37.8 ± 6.1 | 80.0 ± 2.8 | 43.0 ± 4.2 | 63.0 ± 4.2 | - |
| | ER | 42.3 ± 6.9 | 84.0 ± 1.8 | 44.0 ± 5.7 | 58.0 ± 0.0 | - |
| | Medprompt | 41.2 ± 6.2 | 86.0 ± 2.6 | 43.0 ± 1.4 | 54.0 ± 5.7 | - |
| GPT-4(V) | Zero-shot | **70.3** ± 2.0 | 88.7 ± 2.3 | 62.0 ± 2.0 | 67.0 ± 1.4 | 40.0 ± 5.3 |
| | Few-shot | 69.4 ± 1.0 | 86.7 ± 3.1 | 69.0 ± 4.2 | 72.0 ± 2.8 | 35.3 ± 5.0 |
| | + CoT [83] | 72.7 ± 7.7 | 78.0 ± 2.0 | 66.0 ± 5.7 | 70.0 ± 0.0 | 36.2 ± 5.2 |
| | + CoT-SC [82] | 52.1 ± 6.4 | 83.3 ± 3.1 | 68.0 ± 2.8 | **76.0** ± 2.8 | 51.7 ± 4.0 |
| | ER [67] | 61.3 ± 2.4 | 82.7 ± 2.3 | **71.0** ± 1.4 | **76.0** ± 5.7 | 50.0 ± 0.0 |
| | Medprompt [59] | 59.5 ± 17.7 | 87.3 ± 1.2 | 70.7 ± 4.3 | 71.0 ± 1.4 | 53.4 ± 4.3 |
| Gemini-Pro(Vision) | Zero-shot | 49.9 ± 6.5 | 88.9 ± 6.4 | 42.7 ± 3.9 | 40.0 ± 1.5 | 40.0 ± 2.8 |
| | Few-shot | 47.1 ± 5.6 | 89.2 ± 3.4 | 41.0 ± 1.4 | 44.0 ± 3.9 | 39.2 ± 1.2 |
| | + CoT [83] | 65.5 ± 4.9 | 91.9 ± 3.4 | 38.0 ± 1.6 | 52.7 ± 7.1 | 45.2 ± 6.8 |
| | + CoT-SC [82] | 60.3 ± 2.4 | 92.0 ± 1.5 | 46.0 ± 0.0 | 51.0 ± 4.2 | **54.9** ± 3.4 |
| | ER [67] | 46.7 ± 6.9 | 58.5 ± 7.5 | 50.8 ± 5.8 | 53.2 ± 7.8 | 53.2 ± 3.5 |
| | Medprompt [59] | 58.2 ± 5.5 | **92.5** ± 4.5 | 44.4 ± 3.2 | 54.0 ± 5.7 | 51.2 ± 1.9 |

\* **CoT**: Chain-of-Thought, **SC**: Self-Consistency, **ER**: Ensemble Refinement
\* (T): *text*-only, (I): *image+text*, (Y): *video+text*
\* All experiments were tested with 3 random seeds

Table 10: Accuracy (%) on Medical benchmarks with **Group** (🤖🤖) setting. **Bold** represents the best performance for each benchmark and model.

| Category | Method | MedQA (T) | PubMedQA (T) | Path-VQA (I T) | PMC-VQA (I T) | MedVidQA (V T) |
|---|---|---|---|---|---|---|
| | | **Medical Knowledge Retrieval Datasets** | | | | |
| GPT-3.5 | Majority Voting | 60.4 ± 2.1 | 68.5 ± 9.6 | - | - | - |
| | Weighted Voting | 57.3 ± 3.0 | 65.8 ± 11.4 | - | - | - |
| | Borda Count | 55.3 ± 7.1 | 70.2 ± 8.8 | - | - | - |
| | MedAgents [72] | 56.0 ± 5.3 | 55.0 ± 1.4 | - | - | - |
| GPT-4(V) | Majority Voting | 80.6 ± 2.9 | 72.2 ± 6.9 | 56.9 ± 19.7 | 36.8 ± 6.7 | - |
| | Weighted Voting | 78.8 ± 1.1 | 72.2 ± 6.9 | 62.1 ± 13.9 | 25.4 ± 9.0 | - |
| | Borda Count | 70.3 ± 8.5 | 66.9 ± 3.0 | 61.9 ± 8.1 | 27.9 ± 5.3 | - |
| | MedAgents [72] | 79.1 ± 7.4 | 69.7 ± 4.7 | 45.4 ± 8.1 | 39.6 ± 3.0 | - |
| Gemini-Pro(Vision) | Majority Voting | 51.6 ± 2.2 | 65.3 ± 12.9 | 58.2 ± 7.3 | 27.1 ± 5.4 | 50.8 ± 7.4 |
| | Weighted Voting | 52.3 ± 3.3 | 63.7 ± 10.0 | 66.4 ± 11.1 | 20.9 ± 3.8 | 57.8 ± 2.1 |
| | Borda Count | 49.4 ± 9.7 | 57.7 ± 15.0 | **68.2** ± 1.8 | 25.3 ± 8.7 | 54.5 ± 4.7 |
| | MedAgents [72] | 48.4 ± 5.5 | 63.6 ± 6.0 | 64.9 ± 12.5 | 35.1 ± 3.1 | **61.6** ± 4.8 |
| Multi-agent | Reconcile [10] | **81.3** ± 3.0 | **79.7** ± 3.2 | 57.5 ± 3.3 | 31.4 ± 1.2 | - |
| | AutoGen [86] | 60.6 ± 5.0 | 77.3 ± 2.3 | 43.0 ± 8.9 | 37.3 ± 6.1 | - |
| | DyLAN [51] | 64.2 ± 2.3 | 73.6 ± 4.2 | 41.3 ± 1.2 | 34.0 ± 3.5 | - |
| | Meta-Prompting [70] | 80.6 ± 1.2 | 73.3 ± 2.3 | 55.3 ± 2.3 | **42.6** ± 4.2 | - |

| Category | Method | DDXPlus (T) | SymCat (T) | JAMA (T) | MedBullets (T) | MIMIC-CXR (I T) |
|---|---|---|---|---|---|---|
| | | **Clinical Reasoning and Diagnostic Datasets** | | | | |
| GPT-3.5 | Majority Voting | 53.6 ± 2.2 | 83.7 ± 3.3 | 47.0 ± 1.4 | 46.0 ± 4.1 | - |
| | Weighted Voting | 55.2 ± 2.0 | 85.9 ± 3.0 | 49.0 ± 1.4 | 43.0 ± 8.4 | - |
| | Borda Count | 63.9 ± 12.1 | 84.9 ± 1.6 | 50.0 ± 1.0 | 45.0 ± 5.6 | - |
| | MedAgents [72] | 47.3 ± 11.0 | 87.0 ± 4.2 | 41.0 ± 3.3 | 56.0 ± 6.2 | - |
| GPT-4(V) | Majority Voting | 67.8 ± 4.9 | **91.9** ± 2.2 | **70.0** ± 5.7 | 70.0 ± 0.0 | 49.5 ± 10.7 |
| | Weighted Voting | 65.9 ± 3.3 | 90.5 ± 2.9 | 66.1 ± 4.1 | 66.0 ± 5.7 | **53.5** ± 2.2 |
| | Borda Count | 67.1 ± 6.7 | 78.0 ± 11.8 | 61.0 ± 5.6 | 66.0 ± 2.8 | 45.3 ± 6.8 |
| | MedAgents [72] | 62.8 ± 5.6 | 90.0 ± 0.0 | 66.0 ± 5.7 | **77.0** ± 1.4 | 43.3 ± 7.0 |
| Gemini-Pro(Vision) | Majority Voting | 52.3 ± 15.3 | 73.5 ± 4.9 | 47.0 ± 1.4 | 44.0 ± 4.2 | 47.9 ± 6.6 |
| | Weighted Voting | 54.3 ± 16.9 | 64.6 ± 6.5 | 42.0 ± 3.2 | 43.6 ± 3.9 | 43.2 ± 2.0 |
| | Borda Count | 67.0 ± 27.7 | 77.3 ± 9.3 | 37.0 ± 1.8 | 46.1 ± 3.2 | 44.6 ± 4.5 |
| | MedAgents [72] | 43.0 ± 2.7 | 80.5 ± 1.9 | 40.8 ± 2.9 | 50.5 ± 3.6 | 38.7 ± 1.5 |
| Multi-agent | Reconcile [10] | **68.4** ± 7.4 | 90.6 ± 2.5 | 60.7 ± 5.7 | 59.5 ± 8.7 | 33.3 ± 3.4 |
| | AutoGen [86] | 67.3 ± 11.8 | 73.3 ± 3.1 | 64.6 ± 1.2 | 55.3 ± 3.1 | 43.3 ± 4.2 |
| | DyLAN [51] | 56.4 ± 2.9 | 75.3 ± 4.6 | 60.1 ± 3.1 | 57.3 ± 6.1 | 38.7 ± 1.2 |
| | Meta-Prompting [70] | 52.6 ± 6.1 | 77.3 ± 2.3 | 64.7 ± 3.1 | 49.3 ± 1.2 | 42.0 ± 4.0 |

\* **CoT**: Chain-of-Thought, **SC**: Self-Consistency, **ER**: Ensemble Refinement
\* (T): *text*-only, (I): *image+text*, (V): *video+text*
\* All experiments were tested with 3 random seeds

Table 11: Accuracy (%) on Medical benchmarks with **Our** (🤖⋯🤖) method. **Bold** represents the best performance for each benchmark and model.

| Method | MedQA (T) | PubMedQA (T) | Path-VQA (I T) | PMC-VQA (I T) | MedVidQA (V T) |
|---|---|---|---|---|---|
| | **Medical Knowledge Retrieval Datasets** | | | | |
| GPT-3.5 | 64.0 ± 1.6 | 66.0 ± 5.7 | - | - | - |
| GPT-4(V) | **88.7** ± 4.0 | **75.0** ± 1.0 | 65.3 ± 3.9 | 56.4 ± 4.5 | - |
| Gemini-Pro(Vision) | 57.4 ± 1.8 | 71.0 ± 1.6 | **72.0** ± 2.3 | **62.2** ± 7.6 | **56.2** ± 6.7 |

| Method | DDXPlus (T) | SymCat (T) | JAMA (T) | MedBullets (T) | MIMIC-CXR (I T) |
|---|---|---|---|---|---|
| | **Clinical Reasoning and Diagnostic Datasets** | | | | |
| GPT-3.5 | 62.5 ± 6.7 | 85.7 ± 3.0 | 48.6 ± 6.5 | 55.3 ± 4.3 | - |
| GPT-4(V) | **77.9** ± 2.1 | **93.1** ± 1.0 | **70.9** ± 0.3 | **80.8** ± 1.7 | **55.9** ± 9.1 |
| Gemini-Pro(Vision) | 59.2 ± 1.2 | 65.0 ± 6.1 | 47.0 ± 2.2 | 42.9 ± 3.4 | 48.1 ± 5.8 |

\* **CoT**: Chain-of-Thought, **SC**: Self-Consistency, **ER**: Ensemble Refinement
\* (T): *text*-only, (I): *image+text*, (V): *video+text*
\* All experiments were tested with 3 random seeds

Table 12: Accuracy (%) on Medical benchmarks with **Solo/Group/Adaptive** settings with increased number of samples (N=100). All benchmarks except for MedVidQA (`Gemini 1.5 Flash`) were evaluated with `GPT-4o mini`.

| Category | Method | MedQA | PubMedQA | Path-VQA | PMC-VQA | MedVidQA |
|---|---|---|---|---|---|---|
| Single-agent | Zero-shot | 75.0 | 54.0 | 58.0 | 48.0 | 50.0 |
| | Few-shot | 77.0 | 55.0 | 58.0 | 50.0 | 51.0 |
| | + CoT | 78.0 | 50.0 | 59.0 | 52.0 | 53.0 |
| | + CoT-SC | 79.0 | 51.0 | 60.0 | 53.0 | 53.0 |
| | ER | 76.0 | 51.0 | 61.0 | 51.0 | 52.0 |
| | Medprompt | 79.0 | 58.0 | 60.0 | 54.0 | 53.0 |
| Multi-agent (Single-model) | Majority Voting | 79.0 | 68.0 | 63.0 | 52.0 | 54.0 |
| | Weighted Voting | 80.0 | 68.0 | **64.0** | 51.0 | 55.0 |
| | Borda Count | 81.0 | 69.0 | 62.0 | 50.0 | 52.0 |
| | MedAgents | 80.0 | 69.0 | 55.0 | 52.0 | 50.0 |
| | Meta-Prompting | 82.0 | 69.0 | 56.0 | 49.0 | - |
| Multi-agent (Multi-model) | Reconcile | 83.0 | 70.0 | 58.0 | 45.0 | - |
| | AutoGen | 65.0 | 63.0 | 45.0 | 40.0 | - |
| | DyLAN | 68.0 | 67.0 | 42.0 | 48.0 | - |
| Adaptive | **MDAgents (Ours)** | **87.0** | **71.0** | 60.0 | **55.0** | 56.0 |

| Category | Method | DDXPlus | SymCat | JAMA | MedBullets | MIMIC-CXR |
|---|---|---|---|---|---|---|
| Single-agent | Zero-shot | 53.0 | 84.0 | 57.0 | 49.0 | 38.0 |
| | Few-shot | 60.0 | 87.0 | 58.0 | 52.0 | 33.0 |
| | + CoT | 66.0 | 84.0 | 55.0 | 64.0 | 33.0 |
| | + CoT-SC | 68.0 | 84.0 | 57.0 | 60.0 | 40.0 |
| | ER | **76.0** | 80.0 | 56.0 | 59.0 | 43.0 |
| | Medprompt | 70.0 | 84.0 | **62.0** | 60.0 | 43.0 |
| Multi-agent (Single-model) | Majority Voting | 53.0 | 82.0 | 56.0 | 59.0 | 54.0 |
| | Weighted Voting | 52.0 | 86.0 | 56.0 | 56.0 | 52.0 |
| | Borda Count | 53.0 | 86.0 | 56.0 | 59.0 | 51.0 |
| | MedAgents | 56.0 | 80.9 | 51.0 | 58.0 | 40.9 |
| | Meta-Prompting | 53.0 | 79.0 | 56.0 | 51.0 | 48.0 |
| Multi-agent (Multi-model) | Reconcile | 60.0 | 87.0 | 59.0 | 60.0 | 43.3 |
| | AutoGen | 47.0 | 87.0 | 53.0 | 55.0 | 47.0 |
| | DyLAN | 54.0 | 84.0 | 55.0 | 57.0 | 42.0 |
| Adaptive | **MDAgents (Ours)** | 75.0 | **89.0** | 59.0 | **67.0** | 56.0 |

\* **CoT**: Chain-of-Thought, **SC**: Self-Consistency, **ER**: Ensemble Refinement
\* (T): *text-only*, (I T): *image+text*, (V T): *video+text*

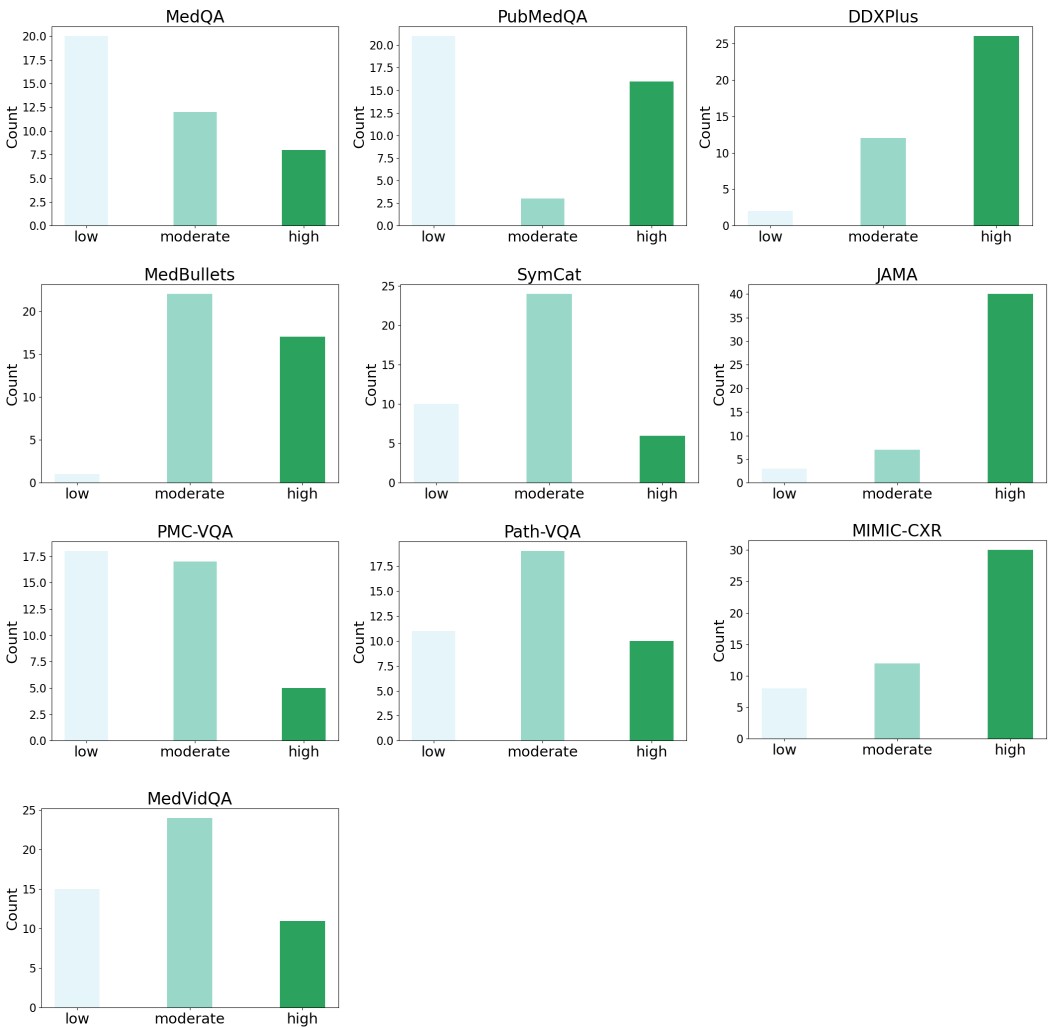

Figure 8: Complexity Distribution for each dataset classified by GPT-4(V) and Gemini-Pro (Vision) (for MedVidQA). The plot illustrates the varying levels of medical complexity across datasets, reflecting the diverse nature of medical question answering, diagnostic reasoning, and medical visual interpretation tasks. For instance, MedQA is categorized under Medical Knowledge Retrieval due to their focus on text-based questions and literature synthesis, while MIMIC-CXR, categorized under Clinical Reasoning and Diagnostic tasks, shows a high complexity distribution due to the need for interpreting detailed radiographic images (See Section in Section 4.1 for the task categorization)

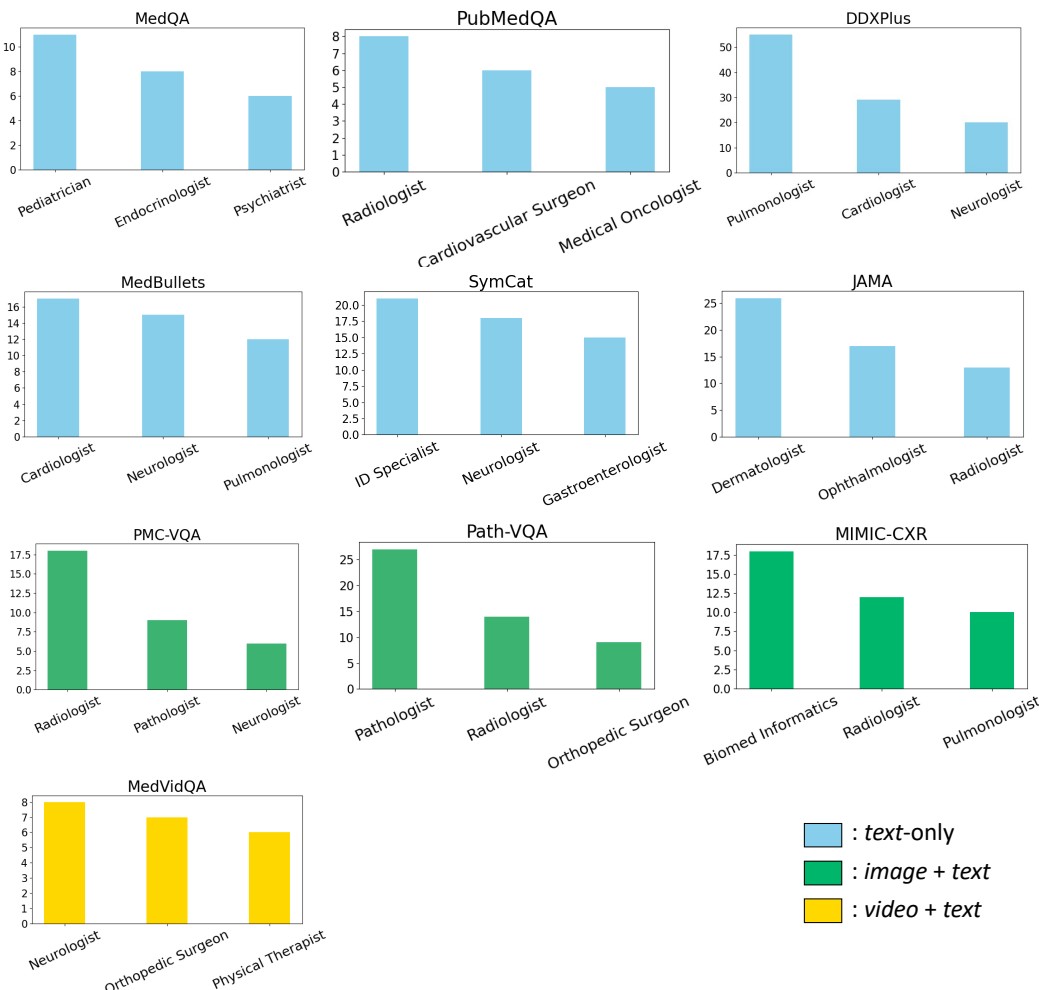

Figure 9: Top-3 most recruited medical experts in each benchmark. The alignment between the dataset characteristics and the recruited experts is evident in several cases. For instance, MIMIC-CXR, which features chest x-ray images, predominantly recruits Radiologists, Pulmonologists, and experts in Biomedical Informatics due to their expertise in interpreting medical imaging.

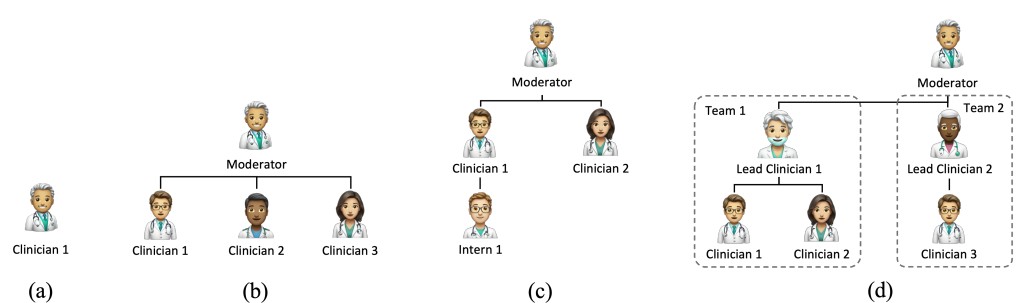

Figure 10: Simplified agent structure examples assigned during the expert recruitment process ranging from (a) A Primary Care Clinician (PCC), (b) Multi-disciplinary Team (MDT), (C) MDT w/ hierarchy to (d) Integrated Care Team (ICT).

**Algorithm 1** Adaptive Medical Decision-making Framework

---

**Require:** Problem $Q$
1:  $Complexity \leftarrow$ COMPLEXITYCHECK$(Q)$                        ▷ Determine the complexity of the medical query
2:  **if** $Complexity = low$ **then**
3:      $Agent \leftarrow$ RECRUIT$(Q, Complexity)$                        ▷ Recruit a Primary Care Clinician agent
4:      $ans \leftarrow Agent(Q)$
5:  **else if** $Complexity = moderate$ **then**
6:      $MDT \leftarrow$ RECRUIT$(Q, Complexity)$                        ▷ Recruit a Multi-disciplinary Team
7:      $Agent \leftarrow$ RECRUIT$(Q, Complexity, MDT)$
8:      $r \leftarrow 0$
9:      $Consensus \leftarrow$ False
10:      $Interaction \leftarrow []$
11:      **while** $r \leq R$ and not $Consensus$ **do**
12:          $Consensus, Log \leftarrow$ COLLABORATIVEDISCUSSION$(Q, MDT)$                        ▷ Iterative discussions
13:          **if** not $Consensus$ **then**
14:              **for all** $Agent \in MDT$ **do**
15:                  $Feedback \leftarrow Moderator(Interaction, Agent)$                        ▷ Review and provide feedback
16:                  $Agent.$UPDATE$(Feedback)$                        ▷ Update the feedback
17:              **end for**
18:              $Interaction \leftarrow Interaction + [Log] + [Feedback]$
19:          **end if**
20:          $r \leftarrow r + 1$
21:      **end while**
22:      $ans \leftarrow Agent(Q, Interaction)$                        ▷ Moderator agent makes the final decision
23:  **else**
24:      $ICT \leftarrow$ RECRUIT$(Q, Complexity)$                        ▷ Recruit an Integrated Care Team
25:      $Reports \leftarrow []$
26:      **for** $Team \in ICT$ **do**
27:          $Report \leftarrow$ GENERATEREPORT$(Q, Team)$                        ▷ Each Team curates a report
28:          $Reports \leftarrow Reports + [Report]$
29:      **end for**
30:      $ans \leftarrow Agent(Q, Reports)$                        ▷ Final decision made
31:  **end if**
32:  **return** $ans$

---

# E   Case Study

## E.1   Medical Decision Making Case Studies

MDM requires efforts of both individual expertise and collaboration to navigate the complexities of patient care. Clinicians often face challenging scenarios that necessitate a comprehensive approach, integrating insights from various specialties to arrive at the best possible outcomes.

### E.1.1   Real-World Medical Cases

Below are the real-world example cases that could be classified as low, moderate, to high complexity cases.

**Case 1: Adjusting Medication Dosage for Chronic Disease (Low Complexity)**   A 55-year-old female patient with type 2 diabetes visits her PCP for a routine check-up. The patient has been taking 500 mg of metformin orally twice a day and has been adhering to a low-carbohydrate diet. Upon testing with fasting glucose level, the glucose level is above normal. PCP reviewed the current medical dosage and increased the dosage to manage the blood glucose level of the patient.

**Case 2: Differential Diagnosis in the Emergency Department (Moderate Complexity)**   A 40-year-old male patient arrives at the emergency department (ED) with a high fever, severe headache, and muscle pain, raising concerns about a potential infectious disease. The ED physician conducts an initial examination but recognizes the need for a more detailed evaluation to identify the underlying cause. The patient is referred to the infectious disease department for further assessment. An infectious disease specialist, along with the ED physician, reviews the patient's symptoms, travel history, and recent exposures. They collaborate on ordering specific diagnostic tests, including blood cultures and imaging studies. Through this teamed decision-making process, they diagnose the patient with dengue fever and promptly initiate appropriate antiviral treatment.

**Case 3: Managing Adverse Responses to Medication in Chronic Disease (High Complexity)**   A 60-year-old female patient with chronic heart failure has been experiencing new symptoms of shortness of breath and mild fever, suggesting either a complication due to her chronic heart failure or a new infection. The urgent care doctor identifies the severity of the situation and promptly refers the patient to the emergency department of a large hospital, where the patient has triaged to see a cardiologist and an infectious disease doctor for specialized care. The team conducts a detailed review of the patient's medication history and current symptoms, does a physical exam to listen to lung sounds, and orders a few exams including labs, a chest x-ray, echocardiogram, and electrocardiogram. The team identifies that the patient has pulmonary effusion and upper respiratory viral infection.

### E.1.2   Medical Cases from MedQA Dataset

Now, let us look at the cases from the MedQA [35] dataset that illustrate either individual PCP or teamed decision-making is crucial in managing medical conditions, ranging from low to high complexity levels of potential cases. These examples highlight the importance of checking the complexity of the case for proper management.

**Case 4: Diagnosis by PCP**   The case below with the "Low Complexity" header is classified as low complexity by a medical doctor. In this case, a PCP can answer this question without consulting a gastroenterologist. The diagnosis of gastric cancer and management based on the manifestation of the disease, that has been described in this question and beyond should be from a gastroenterologist. However, PCPs are expected to have the basic scientific and pathophysiological knowledge that is related to gastric cancer and use that knowledge to solve this problem.

**Case 5: Diagnosis and management by single Pediatric Endocrinologist**   The case below with the "Moderate Complexity" header is classified as moderate complexity by the medical doctor. In this case, a pediatric endocrinologist (specialist) alone can diagnose a patient and have a treatment plan. Note that this patient could have been referred to this pediatric endocrinologist by a PCP who is regularly seeing this patient.

> **Low Complexity**
>
> Question: A 70-year-old man comes to the physician because of a 4-month history of epigastric pain, nausea, and weakness. He has smoked one pack of cigarettes daily for 50 years and drinks one alcoholic beverage daily. He appears emaciated. He is 175 cm (5 ft 9 in) tall and weighs 47 kg (103 lb); BMI is 15 kg/m². He is diagnosed with gastric cancer. Which of the following cytokines is the most likely direct cause of this patient's examination findings?
>
> Answer:
>
> A) TGF-$\beta$
>
> B) IL-6
>
> C) IL-2
>
> **D) TNF-$\beta$**

> **Moderate Complexity**
>
> Question: A 5-year-old girl is brought to the clinic by her mother for excessive hair growth. Her mother reports that for the past 2 months she has noticed hair at the axillary and pubic areas. She denies any family history of precocious puberty and reports that her daughter has been relatively healthy with an uncomplicated birth history. She denies any recent illnesses, weight change, fever, vaginal bleeding, pain, or medication use. Physical examination demonstrates Tanner stage 4 development. A pelvic ultrasound shows an ovarian mass. Laboratory studies demonstrate an elevated level of estrogen. What is the most likely diagnosis?
>
> Answer:
>
> **A) Granulosa cell tumor**
>
> B) Idiopathic precocious puberty
>
> C) McCune-Albright syndrome
>
> D) Sertoli-Leydig tumor

**Case 6: Diagnosis and management by multidisciplinary team** The case below is of a "High Complexity" patient primarily having neurological symptoms but with problems with vision, which requires a neurologist to consult to ophthalmology department for further evaluation.

> **High Complexity**
>
> Question: A 63-year-old woman presents to her primary-care doctor for a 2-month history of vision changes, specifically citing the gradual onset of double vision. Her double vision is present all the time and does not get better or worse throughout the day. She has also noticed that she has a hard time keeping her right eye open, and her right eyelid looks 'droopy' in the mirror. Physical exam findings during primary gaze are shown in the photo. Her right pupil is 6 mm and poorly reactive to light. The rest of her neurologic exam is unremarkable. Laboratory studies show an Hb A1c of 5.0%. Which of the following is the next best test for this patient?
>
> Answer:
>
> A) Direct fundoscopy
>
> B) Intraocular pressures
>
> **C) MR angiography of the head**
>
> D) Temporal artery biopsy

## E.2 Cases Studies with MDAgents

In this section, we provide two examples of our framework with *moderate* (Figure 11) and *high* (Figure 12) complexity in PMC-VQA (*image+text*) and DDXPlus (*text-only*) respectively. These

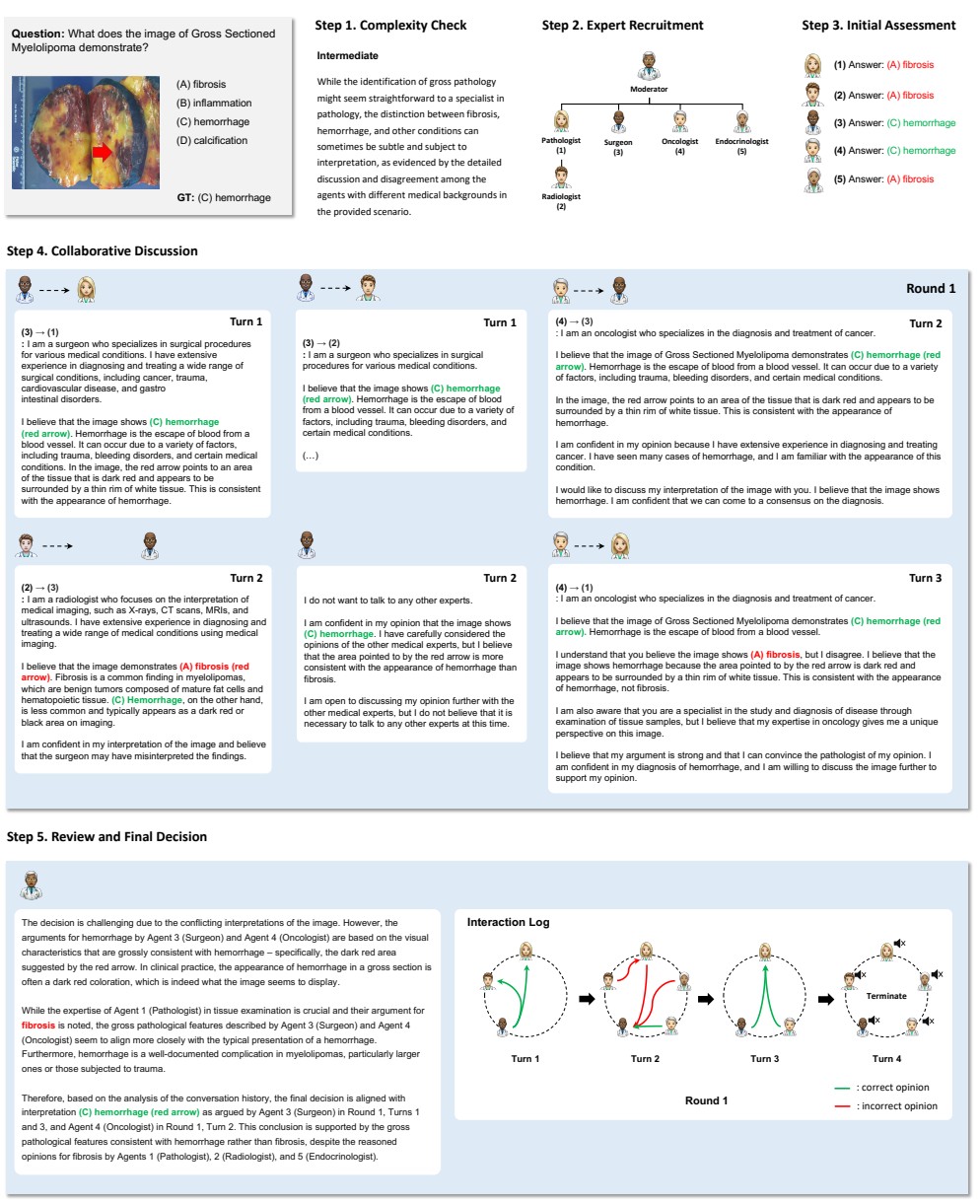

Figure 11: Illustration of our proposed framework in *moderate* complexity setting. Given a medical query (*image + text*) the framework performs reasoning in five steps: (i) complexity check, (ii) expert recruitment, (iii) initial assessment, (iv) collaborative discussion, and (v) review and final decision-making. Green text represents the correct answer and the Red text represents the incorrect answer.

case studies reveals how our framework provides an environment for agents to collaborate, gather information, moderate and make final decisions in complex medical scenarios.

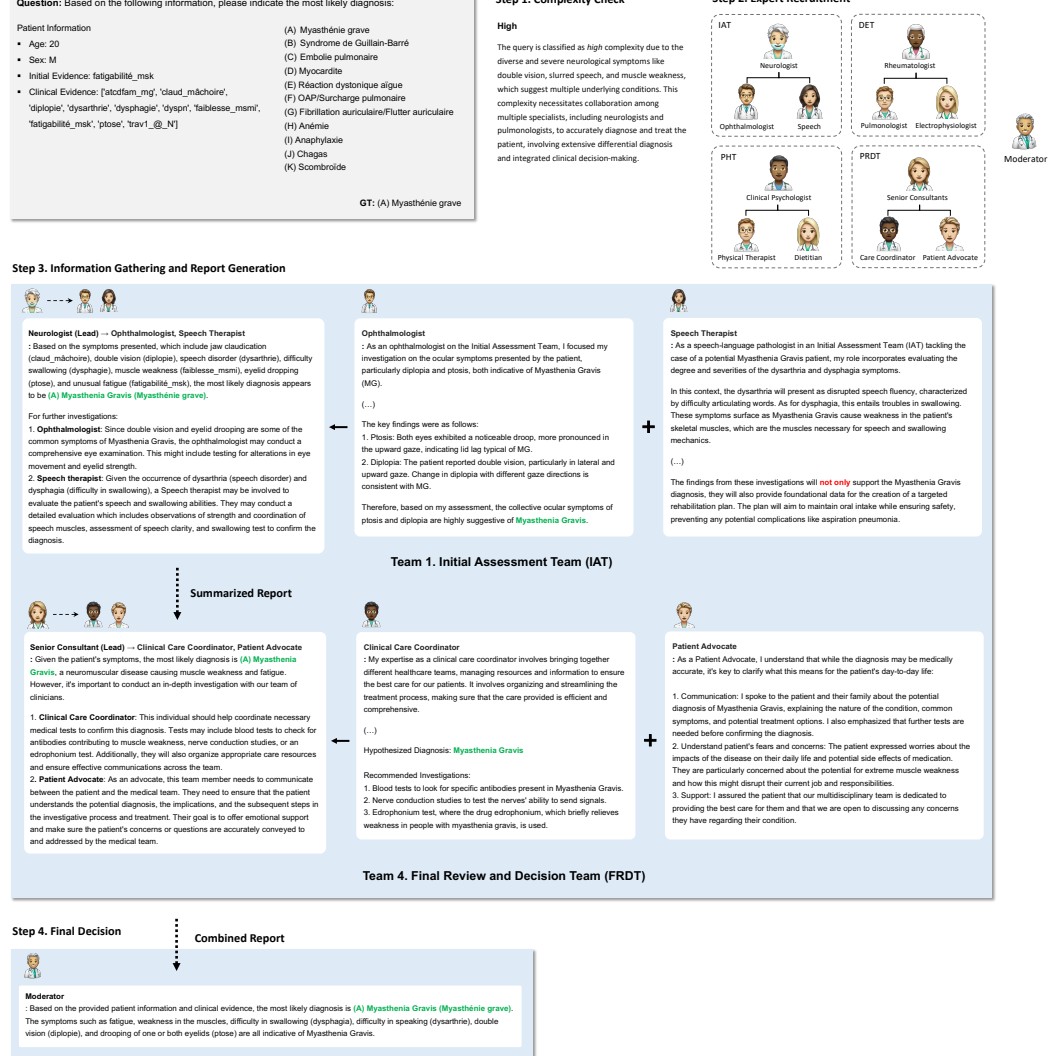

Figure 12: Illustration of our proposed framework in *high* complexity setting. Given a medical query (*text*-only) the framework performs reasoning in four steps: (i) complexity check, (ii) expert recruitment, (iii) information gather and report generation, (iv) final decision. Green text represents the correct answer.

# F  Medical Complexity Comparison with Human Physicians

The core premise of our MDAgent framework is its ability to adapt to the complexity of medical tasks. To validate this approach and gain insights into how LLMs perceive medical complexity compared to human experts, we conducted an annotation study. This study aimed to explore the alignment between LLMs and physicians in assessing medical question complexity, a critical factor in the effectiveness of our MDAgent framework.

**Study Design**  We selected 50 representative questions from the MedQA dataset, ensuring a balanced representation across USMLE steps 1, 2, and 3. This selection process aimed to cover a wide range of medical topics and complexity levels, mirroring the diverse challenges that our MDAgent framework is designed to address.

Three physicians participated in our study: two with two years of Internal Medicine training (Post Graduate Year 2, PGY-2) and one general physician. This composition allowed us to capture a range

of clinical perspectives. The physicians rated each question on a scale of -1 (low complexity), 0 (moderate complexity), and 1 (high complexity).

**Inter-rater Reliability**  To quantify the agreement among our physician raters, we employed Intraclass Correlation Coefficients (ICC). ICC is a widely used statistical measure in medical research for assessing the consistency of ratings among multiple raters. We specifically chose two ICC variants:

- ICC2k (Two-way random effects, average measures): 0.269 [-0.14, 0.55]
- ICC3k (Two-way mixed effects, average measures): 0.280 [-0.15, 0.57]

ICC2k was selected because it assumes our raters are randomly selected from a larger population of similar raters, allowing for generalization of our findings. ICC3k, on the other hand, treats the raters as fixed, focusing on the consistency among our specific set of physicians.

Both ICC values indicate moderate agreement among the raters. This level of agreement reflects the inherent complexity and subjectivity in evaluating medical questions, even among trained professionals. It also highlights the challenging nature of the task our MDAgent framework aims to address.

**Annotation Interface**  To facilitate the annotation process for both physicians and LLMs, we developed a specialized interface. This interface was designed to present medical questions in a clear and consistent manner, allowing for efficient and standardized complexity ratings. Figure 13 shows a screenshot of the annotation interface used in our study.

Figure 13: Annotation interface used for medical complexity assessment. The latest version can be found at `https://dxagents.github.io/2024/05/01/medqa.html`.

**LLM Annotations and Comparison**  To compare LLM performance with human expert judgments, we employed several state-of-the-art models to annotate the same set of questions. We then compared these assessments with the majority opinion of the physicians, determined by the mode of their ratings (or the mean in cases of complete disagreement).

Table 13 presents the Pearson correlation between each LLM's complexity ratings and the physicians' majority opinions:

| Model | Correlation with Physician Majority |
|---|---|
| gpt-4o-mini | -0.090 |
| gpt-4o | 0.022 |
| gpt-4 | 0.070 |
| gemini-1.5-flash | 0.110 |

Table 13: Correlation between LLM complexity ratings and physician majority opinions

The results of our study provide valuable insights into the current state of LLM capabilities in medical complexity assessment and underscore the importance of our MDAgent framework:

1. **Subjectivity in medical complexity**: The moderate ICC values among physicians highlight the inherent subjectivity in assessing medical question complexity. This finding validates our approach in MDAgent, which doesn't rely on a single, fixed assessment of complexity but rather adapts its collaboration structure dynamically.

2. **Current LLM limitations**: The low correlations between LLM and human assessments indicate that current LLMs may not fully capture the nuances that human experts consider when evaluating medical complexity. This observation reinforces the need for our MDAgent framework, which can compensate for individual LLM limitations through collaborative decision-making.

3. **Potential for improvement**: The variation in correlation across different LLM models (from -0.090 to 0.110) suggests there is room for improvement in LLM performance. This aligns with our MDAgent approach, which can leverage the strengths of multiple models and adapt to future improvements in LLM capabilities.

4. **Value of human expertise**: The discrepancy between LLM and physician assessments underscores the continued importance of human medical expertise. Our MDAgent framework acknowledges this by incorporating human-like collaboration strategies and the potential for human oversight in critical decisions.

5. **Adaptability of MDAgent**: The challenges revealed in this study highlight the wisdom of our MDAgent's adaptive approach. By dynamically adjusting its collaboration structure based on perceived task complexity, MDAgent can mitigate the limitations of individual LLMs and approach the nuanced understanding demonstrated by human experts.

