# OpenReview forum: "MDAgents: An Adaptive Collaboration of LLMs for Medical Decision-Making"
_NeurIPS.cc/2024/Conference — NeurIPS 2024 oral_

### Official Review · Reviewer_QC4Q · 2024-06-14

**Soundness:** 3
**Presentation:** 3
**Contribution:** 3
**Rating:** 7
**Confidence:** 5

**Summary:**

This paper proposes MDAgents (Medical Decision-making Agents), a LLM collaboration framework for medical question answering. Given a single-modal or multi-modal medical question, MDAgents first classifies its complexity into low, moderate, and high. Based on the complexity checking result, MDAgents assigns a single primary care clinician LLM agent (for low complexity), a team of multidisciplinary LLM agents (for moderate complexity), or a team of integrated care LLM agent (for high complexity). The agent collaboration adopts multi-turn discussion and iterative report refinement. Evaluated on multiple medical QA datasets, MDAgents show better performance than a variety of baseline models, including other prompting strategies as well as other agent framework. Overall, this is an interesting paper.

**Strengths:**

1. The writing is generally clear and the displays are informative.
2. A new medical domain-specific agent collaboration method has been proposed for decision making.
3. Comprehensive experiments have been conducted to show the superior performance of MDAgents.
4. Interesting additional analysis.

**Weaknesses:**

1. The main issue of this article is that the reported scores are not consistent with the literature, so it is unclear whether MDAgents is really state-of-the-art. For example, the original Medprompt paper reported their performance on MedQA as 90.2 and PubMedQA as 82.0. However, this paper reports Medprompt scores of 82.4 and 51.8 for these two datasets. The authors need to explain such discrepancy.
2. The complexity checker is something novel but is not well evaluated. The authors might need to sample a set of questions and ask human physicians to score its complexity (e.g., from 1-5), and report the correlation between LLM complexity and human complexity.
3. Some figures in the results sections are confusing. I am not exactly sure what Figure 3 means, and it contains additional lines that should be removed. Additionally, for figure 5, does the "Low" mean the subset performance of questions classified as "Low", or the performance if all questions are classified as "Low"?
4. Studies on other medical agents (e.g., https://arxiv.org/abs/2402.13225) should also be discussed.

**Questions:**

1. What's the relative cost of MDAgents v.s. GPT-4 zero-shot CoT on each of the dataset?
2. If you remove the LLM complexity checker, and use the ICT method for all questions, will it achieve the highest result?
3. Can you aggregate the GPT-3.5 results like the main results table? They are currently scattered in different tables.

**Limitations:**

1. The evaluations, while comprehensive, are mostly on multi-choice question answering tasks. This is not a realistic setting in medicine.

---

> ### Author Rebuttal · Authors · 2024-08-07
>
> We sincerely appreciate your thoughtful review and the valuable insights you provided. Your feedback helps us clarify key aspects of our research and improve the overall quality of our submission.
>
> **W1. The reported scores not consistent with the literature**
>
> Thank you for pointing out the discrepancies in the reported scores. We appreciate your attention to this detail and would like to clarify the differences between our experimental setup and the original paper [1] that introduced Medprompt:
>
> **1) Dataset Differences:** The high accuracy reported by [1] for Medprompt was achieved using the MedQA-US dataset with 4-option questions. Our evaluation, however, was conducted on the MedQA-US dataset with 5-option questions, which is inherently more challenging and likely contributes to the lower accuracy.
>
> **2) Implementation Variations:** In the original Medprompt implementation, they utilized five kNN-selected few-shot exemplars with a 5x ensemble. For our experiments, we used three exemplars to ensure fair comparisons with other methods in our main experiments. We initially considered using a different number of exemplars to calibrate our implementations further. However, we chose a smaller number due to the increased cost and time required for comprehensive testing. Under similar conditions, such as using more exemplars like [1], we anticipate that our MDAgents and other baseline methods might also demonstrate improved performance.
> We will ensure these distinctions in the implementations are clearly outlined in our paper to provide proper context for the reported results.
>
> **W2. What is the correlation between LLM complexity scores and human Physician's judgments?**
>
> We have addressed this in the general response by detailing a study where three human physicians annotated question complexity and conducted a correlation analysis with LLM assessments.
>
> **W3 & Q2. Figure 3 and 5 confusing and what if we remove LLM complexity checker in the ablation?**
>
> For Figure 5, "Low," "Moderate," and "High" denote the performance outcomes when all questions in the dataset are manually set into each respective complexity category, rather than relying on the complexity obtained by the LLM complexity checker. This means:
>
> * **"Low"** shows the accuracy when all questions are set to low complexity.
> * **"Moderate"** indicates the accuracy when all questions are set to moderate complexity.
> * **"High"** reflects the accuracy when all questions are set to high complexity.
>
> This setup was designed to evaluate how the model's performance varies when operating under uniform complexity assumptions across the dataset. We will ensure that the figure description and ablation setup in our paper (Section 4.3) are updated to clearly explain this methodology.
>
> **W4. Studies on other medical agents should also be discussed**
>
> We will include a discussion of the study [1] suggested and others [2,3,4,5] to better contextualize our work and clarify how MDAgents compare with existing approaches in medical decision-making.
>
> **Q1. What is the relative cost of MDAgents vs. gpt-4 zero-shot CoT on each dataset**
>
> As detailed in Table 3 of the attached pdf file, MDAgents require higher costs across the datasets compared to Zero-shot CoT. Our methods use a 3-shot setting across different medical complexities and recruit multiple agents, which are needed to effectively handle the complexity of medical datasets which contributes to the enhanced performance.
>
> **Q3. Need to aggregate GPT-3.5 results in Table 3**
>
> We will aggregate the GPT-3.5 results into the main results table for clarity and easier comparison with other models.
>
> **L1. Evaluations limited to multi-choice question tasks which is not a realistic medical setting**
>
> To address the issue, we conducted additional experiments using the MedQA dataset without predefined options, aiming to better mirror the open-ended nature of clinical decision-making.
>
> Recognizing that real-world medical scenarios often involve complex, multi-turn interactions, we are committed to refining our evaluation methods to more accurately simulate actual clinical conditions. In our latest experiments with the gpt-4o-mini model, we evaluated our method alongside 3-shot CoT-SC and Reconcile, using 100 samples from the MedQA dataset in an open-ended format. The results were as follows:
>
> * **3-shot CoT-SC:** 52 % accuracy
> * **Reconcile:** 40 % accuracy
> * **Ours:** 56 % accuracy
>
> These results indicate that our approach performs competitively even in more realistic settings, underscoring its potential in clinical applications. We are open to further suggestions and welcome any recommendations for additional datasets or evaluation frameworks that could enhance the realism and robustness of our assessments.
>
> We believe our responses have addressed your concerns and provided clarity. Please let us know if you require any additional information or further clarifications for your re-evaluation.
>
> **References**
>
> [1] Nori, H., Lee, Y. T., Zhang, S., Carignan, D., Edgar, R., Fusi, N., ... & Horvitz, E. (2023). Can generalist foundation models outcompete special-purpose tuning? case study in medicine.
>
> [2] Jin, Q., Wang, Z., Yang, Y., Zhu, Q., Wright, D., Huang, T., … & Lu, Z. (2024). AgentMD: Empowering Language Agents for Risk Prediction with Large-Scale Clinical Tool Learning.
>
> [3] Li, J., Wang, S., Zhang, M., Li, W., Lai, Y., Kang, X., ... & Liu, Y. (2024). Agent hospital: A simulacrum of hospital with evolvable medical agents.
>
> [4] Fan, Z., Tang, J., Chen, W., Wang, S., Wei, Z., Xi, J., ... & Zhou, J. (2024). Ai hospital: Interactive evaluation and collaboration of llms as intern doctors for clinical diagnosis.
>
> [5] Yan, W., Liu, H., Wu, T., Chen, Q., Wang, W., Chai, H., ... & Zhu, L. (2024). ClinicalLab: Aligning Agents for Multi-Departmental Clinical Diagnostics in the Real World.

---

> > ### Comment · Reviewer_QC4Q · 2024-08-08
> >
> > Thank you for your response. I have increased my score.

---

### Official Review · Reviewer_mJeV · 2024-07-09

**Soundness:** 2
**Presentation:** 3
**Contribution:** 3
**Rating:** 6
**Confidence:** 3

**Summary:**

This paper introduces a framework called MDAgents, which optimizes collaboration between multiple large language models (LLMs) for medical decision-making tasks. The main technical contribution of MDAgents is the deployment of a moderator agent to assess the complexity of incoming queries, categorizing them into low, moderate, and high difficulty. To improve efficiency, MDAgents either call single agents to solve low-complexity problems or use several agents to work together using real health studies-inspired collaboration schemes.

**Strengths:**

- Comprehensive experiments and benchmarking.
- Performance appears promising.

**Weaknesses:**

- The complexity assessment lacks details. More explanation on "low, moderate, high" is required.
- Additionally, the assessment is entirely determined by an LLM, raising concerns about whether poor judgment by the moderator agent may propagate.
- The experiments employing just 50 samples per dataset, may not adequately represent the performance of the proposed methods.

**Questions:**

- MedAgents seems to use the same LLMs for all the agents, with the differences among the agents being the Agent Initialization Prompts. When the Multi-disciplinary Teams were recruited, is assigning an agent a role enough to let the LLM act like an expert in that specific discipline, which requires a lot of domain knowledge? How about equipping different agents with different knowledge for RAG?
- The observation that questions labeled as "Low" complexity have lower accuracy rates than "Moderate" ones in Figure 3 casts doubt on the reliability of the moderator's assessment. Also, the reasoning claim that the complexity assessment can increase accuracy by at least 80% is way more optimistic and seems like a really bold statement to me.
- In the ablation study, assigning all queries the highest complexity level results in reduced accuracy compared to the adaptive approach. Why is that? Does it mean calling multiple agents to collaborate in a complicated might not always lead to better outcomes, and incur higher API calls?

**Limitations:**

Addressed

---

> ### Author Rebuttal · Authors · 2024-08-07
>
> We appreciate your detailed review and the opportunity to refine our work based on your feedback. Your suggestions are crucial in guiding our efforts to provide a more comprehensive analysis and evaluation.
>
> **W1 & W2. The complexity assessment lacks details and concerns about judgment made by the moderator agent may propagate.**
>
> We address this issue in Section 4.3 through an ablation study, the results of which are presented in Figure 5. This study evaluates the effectiveness of our adaptive complexity selection mechanism against static assignments across different modalities.
>
> Our findings indicate that the adaptive method significantly outperforms static settings, demonstrating robustness in our approach to complexity assessment and reducing the potential for error propagation. Additionally, a detailed comparison of human doctor annotations with the LLM’s assessments is provided in the pdf file attached.
>
> **W3. 50 Samples per dataset may not show true method performance**
>
> Please refer to the general response and the pdf file attached for extra experiments with N=100 samples for all datasets and with entire test samples for the MedQA dataset.
>
> **Q1. Is assigning an agent a role enough for LLMs to act like experts? What if equipping different knowledge with RAG?**
>
> Thank you for your idea about the knowledge initialization for agents using RAG in our MDAgents framework. To address your point on domain expertise, we conducted extra experiments on top of Table 3:
>
> MDAgents: 71.8%
>
>  \+ MedRAG: 75.2%
>
>  \+ Medical Knowledge Initialization: 76.0% (**your suggestion**)
>
>  \+ Moderator’s Review: 77.6%
>
>  \+ Moderator’s Review & MedRAG: 80.3%
>
> These results indicate that augmenting agents with specific knowledge and structured reviews have potential to improve their ability to simulate domain expertise. We will detail these findings in our revised manuscript.
>
> **Q2. Doubts on Complexity Labels and Optimistic Accuracy Claims in Figure 3**
>
> To re-explain Figure 3, we need to explain why this experiment is needed, which will help to understand the meaning of Figure 3.
>
> It is important to accurately assign difficulty levels to medical questions. For instance, if a medical question is obviously easy, utilizing a team of specialists (such as an Interdisciplinary Doctor Team, IDT) might be excessive and potentially lead to overly pessimistic approaches. Conversely, if a difficult medical question is only tackled by a PCP, the problem might not be adequately addressed.
>
> The core issue here is the LLM's capability to classify the difficulty of medical questions appropriately. If an LLM inaccurately classifies the difficulty level, the chosen medical solution may not be suitable, potentially leading to the wrong decision making. Therefore, understanding what constitutes an appropriate difficulty level is essential.
>
> We hypothesize that the appropriate difficulty for each question corresponds to the difficulty level at which the probability of correctly solving the question is highest, as we cannot incorporate doctor in the difficulty decision making (while we also have ablation studies with human doctors as well)
>
> To determine this, we assessed the accuracy of solutions across various difficulty levels. Specifically, we evaluated 10 medical problems (increased to 25 after rebuttals) by solving each problem 10 times at each difficulty level. By measuring the success rate, we aimed to identify the difficulty level that yielded the highest accuracy.
>
> This rigorous approach ensures that the LLM's classification of problem difficulty aligns with the most effective and accurate medical solutions, thereby optimizing the application of medical expertise to each question.
>
> Going back to Reviewer mJeV’s question,
> >The observation that questions labeled as "Low" complexity have lower accuracy rates than "Moderate" ones in Figure 3 casts doubt on the reliability of the moderator's assessment.
>
> It is important to clarify that Figure 3(b) does not indicate specific questions labeled as "Low" complexity. Instead, it shows the probability that the LLM can correctly answer questions when c our “Low Complexity” solution is applied for **all questions**. This explanation extends to Figures 3(c) and 3(d) as well. Figure 3(a), on the other hand, illustrates whether the LLM is choosing the difficulty level that provides the highest accuracy.
>
> > Also, the reasoning claim that the complexity assessment can increase accuracy by at least 80% is way more optimistic and seems like a really bold statement to me.
>
> We did not make such a claim. Our assertion is that the LLM is automatically selecting the appropriate difficulty level with an accuracy rate close to 80%.
>
> **Q3. Why does assigning high complexity to all queries reduce accuracy and increase API costs?**
>
> To address this issue, we conducted additional experiments to see if we missed the benefit to improve the performance in high complexity cases.
>
> For the image+text scenario, we explored various collaborative settings and found these outcomes:
>
> * Sequential & No Discussion: 39.0%
>
> * Sequential & Discussion: 45.0%
>
> * Parallel & No Discussion: 56.0%
>
> * Parallel & Discussion: 59.0%
>
> This indicates the importance of multi-turn discussions, particularly in complex cases and the exclusion of this feature likely contributed to lower performance.
>
> We hope our detailed responses have addressed your concerns effectively. Please feel free to add follow-up questions for further clarifications or updates needed for your re-evaluation.

---

> ### Comment · Reviewer_mJeV · 2024-08-12
>
> Thanks for the response. I will increase my score to acknowledge the authors' efforts in addressing my questions.

---

### Official Review · Reviewer_mREC · 2024-07-10

**Soundness:** 3
**Presentation:** 3
**Contribution:** 3
**Rating:** 5
**Confidence:** 3

**Summary:**

In this paper, the authors propose MDAgent, a multi-agent framework for medical decision making. In this framework, the complexity of the problem is initially assessed by an agent. Based on this assessment, either a single agent or a group of agents is assigned to solve the problem. The authors evaluate their framework on 10 medical benchmarks, including both text-only and multi-modal datasets. Experimental results show that MDAgent outperforms existing baselines on 7 datasets and achieves high efficiency compared to other multi-agent methods.

**Strengths:**

- The authors propose a adaptive multi-agent framework for medical decision making, which dynamically assesses the complexity of each problem and assigns na appropriate group of agents to solve it.
- The authors conduct experiments on 10 datasets, including both text-only and multi-modal ones, and compare their method with various baselines. Experimental results show that the proposed method outperforms existing baselines on 7 datasets.
- The authors focus not only on performance, but also on efficiency and robustness, which are crucial for realistic application.

**Weaknesses:**

- The description of MDT and ICT is unclear. For example, how are the prompts for each specialist prepared (the {{description}} in Agent initialization prompt, Appendix C.2), are they handcrafted or generated from LLM? How does the hierarchy shown in Figure 10 (c) and (d) work?
- The experiment shown in Figure 3 uses only 10 questions from a single dataset, which is not convincing. Additionally, subfigures (b), (c), and (d) are not mentioned in the analysis. What conclusions can be drawn from these results?
- The results shown in Figure 5 seem counterintuitive. For image + text and video + text, the score for low is higher than for moderate and high. This suggests that some questions that can be correctly solved with a single agent result in incorrect answers when multiple agents are involved. More analysis is needed to uncover the reason.
- The experiment incorporating the moderator's review and RAG shown in Table 3 has little relation to other parts of the paper. Considering that the moderator's review and RAG can also be combined with other multi-agent methods, the authors should compare the performance gain when attaching the moderator's review and RAG with different multi-agent frameworks if they want to demonstrate that their method is more suitable for the moderator's review and RAG.

**Questions:**

- The authors could discuss more about the motivation behind the design of PCC, MDT, and ICT, considering that readers may not be familiar with the medical decision-making process. For example, why does MDT contain a multi-turn discussion but not ICT? Shouldn't the most complex questions require more discussion between agents?
- In section 4.3, the authors discuss Figure 5 in the first paragraph and Table 3 in the second paragraph. However, Table 3 appears on page 8 before Figure 5 appears on page 9, which could confuse readers.
- The formatting instructions for NeurIPS 2024 state that "All tables must be centered, neat, clean, and legible. The table number and title always appear before the table." However, Table 3 violates this rule.

**Limitations:**

- Most of the datasets used for evaluation are in multiple-choice question or true/false question format. In real-world scenarios, there are no options for doctors to choose from. Therefore, the authors could include some open-ended questions to simulate realistic applications.

---

> ### Author Rebuttal · Authors · 2024-08-07
>
> We appreciate your review and the opportunity to address your concerns. We have conducted numerous additional experiments to validate our approach and enhance the robustness of our findings.
>
> **W1. The description of MDT and ICT is unclear.**
>
> * **Agent Initialization:** In our framework, the roles and descriptions for specialists within the MDT and ICT are dynamically determined by a specialized recruiter LLM. This process automates the generation of role-specific prompts, ensuring each agent is precisely tailored to the demands of the case.
>
> * **Hierarchy Explanation:** The hierarchical configuration in MDT and ICT, as depicted in Figure 10 (c-d), is inspired by traditional clinical reporting structures. This protocol ensures structured communication flow and oversight, preventing information discrepancies and fostering coherent team collaboration.
>
> **W2. Experiment shown in Figure 3 with only 10 questions not convincing**
>
> We acknowledge the limitation of using only 10 questions in Figure 3. Initially, we had 10 questions because this already required generating 300 question-answer pairs (10 solutions * 3 difficulty levels * 10 questions). However, with the current significant reduction in price offered by gpt-4o-mini, we added 15 more questions which results in 750 question-answer pairs.
>
> We will expand our experiments to include more questions from multiple datasets to provide a more robust evaluation.
>
> **W3. The results in Figure 5 seem counterintuitive**
>
> In response to your concerns, we have conducted additional experiments and analyses to clarify these outcomes.
>
> For the image+text scenario, further experiments with different settings has revealed the following results:
>
> * Sequential & No Discussion: 39.0%
> * Sequential & Discussion: 45.0%
> * Parallel & No Discussion: 56.0%
> * Parallel & Discussion: 59.0%
>
> These results suggest that the integration of multi-turn discussions substantially benefits the decision-making process, particularly in complex cases. The initial absence of this feature in our methodology likely contributed to the earlier lower performance figures.
>
> For the video+text scenario, the use of the deprecated gemini-pro vision model initially restricted multi-turn chats (multi-modal cases). By summarizing the video content with one agent and then re-initializing another for further multi-turn discussions, we attempted to overcome this limitation. We assume that this approach might have influenced the accuracy in the moderate and high complexity scenarios.
>
> To thoroughly address these issues and refine our understanding, we plan to further conduct detailed experiments focused on:
>
> * Investigating the impact of different agent initialization strategies on the consistency and accuracy of outcomes (with gemini-1.5 flash).
>
> We will ensure that these additional analyses are included in the revised manuscript for the camera-ready version, aiming to provide a more comprehensive understanding of the interplay between model capabilities and task complexities.
>
> **W4. Need to Compare Moderator's Review and RAG Integration with Other Multi-Agent Frameworks to Demonstrate Suitability**
>
> Our primary focus was on demonstrating how the MDAgents framework can enhance medical decision-making through adaptive collaboration structures with initial complexity classification. The inclusion of the moderator's review and Retrieval-Augmented Generation (RAG) in Table 3 was intended to show the potential improvements in accuracy when integrating external knowledge sources and structured review processes.
>
> While we recognize that the moderator's review and RAG could be applied to other multi-agent frameworks, our aim was to illustrate their effectiveness specifically within MDAgents. However, it's important to note that RAG itself is not a medical-specific technique. We focused on demonstrating how the integration of RAG and structured reviews can be particularly effective within a medical-aware structured adaptive multi-agent system compared to a naive multi-agent system.
>
> **Q1. Discuss more about the motivation behind the design of PCP, MDT, and ICT, and why doesn't ICT include multi-turn discussions?**
>
> The design of PCP, MDT and ICT represents the real-world clinical decision making processes which is mostly dependent on the complexity of the medical cases (refer to Appendix Section D.1.1. for real-world examples). If the case or task is with low complexity where PCP could solve it without consulting specialists (Case 1, Appendix D.1.1), if it is moderate complexity PCP might have to consult to a specialist agents (Case 2, Appendix D.1.1.), and lastly if the case is complex so that it involves multi-disciplinary consult (Case 3, Appendix D.1.1) we let multidisciplinary agents interact with one another.
>
> **Additional experimental results with ICT setting (Accuracy):**
>
> * Sequential report generation w/ discussion: 84%
> * Sequential report generation w/o discussion: 78% (Our previous approach)
> * Parallel report generation w/ discussion: 82%
> * Parallel report generation w/o discussion: 80%
>
> The results indicate that incorporating discussions among lead clinicians in ICT enhances decision-making accuracy, particularly in sequential report generation. This evidence supports your point that complex cases benefit from more extensive deliberation.
>
> In the updated manuscript, we will adjust the ICT model to include more robust discussion protocols.
>
> **Q2 & Q3. The order of Table 3 Appearing Before Figure 5 confuses readers and Table 3 violates NeurIPS 2024 formatting rules**
>
> In the final version, we will ensure that the figures and tables are presented in the same order as they are discussed in the text to enhance clarity for the readers and revise it to comply with the guidelines in the updated manuscript.
>
> We believe our extensive additional experiments and clarifications have addressed your queries. Please let us know if further details are required to support the re-evaluation of our work.

---

> > ### Comment · Reviewer_mREC · 2024-08-12
> >
> > Thank you for your reply. I have updated my score.

---

### Official Review · Reviewer_g9qW · 2024-07-13

**Soundness:** 3
**Presentation:** 3
**Contribution:** 3
**Rating:** 7
**Confidence:** 3

**Summary:**

The paper presents MDAgents, a multi-agent LLM system for answering medical questions, ranging from medical question answering and diagnostic reasoning to medical visual interpretation. The main novelty is a crafted collaboration scheme of multiple agents with designated roles, where a medical question is categorized into being of low, medim or high complexity. The question is then delegated to either individual specialized LLMs or teams of LMMs, and finally a decision is taken by a last LLM. The complete system does not focus on training or fine-tuning specialized medical agents, but rather use an appropriate foundational model and designated prompts to design the individual agents.

The approach is evaluated on ten medical diagnosis datasets, where 50 samples are used for testing. The authors compare their adaptive multi-agent system with a solo agent as well as a fixed group of agents, each with respective SoTA approaches. The results are based on GPT-4(V) or Gemini-Pro(Vision). The results show that MDAgents is either on-par or better than its competitors for all datasets.

**Strengths:**

- Important use case: LLMs for medical decision making have large potential to add value, as they might support physicians or directly patients.
- Novel idea: The paper makes a good case in comparing the main idea to relevant related works, emphasizing how the combination of used concepts and their grouding in the respective agent roles is new. The related work section is sufficiently structured and broad and shows how the current work extends prior work.
- Good evaluation setup in terms of used datasets and competitors: The chosen dataset range is sensible and all relevant competitors, as discussed in the related work section, seem to be evaluated against.
- Strong results: The approach beats the other SoTA approaches in 70% of used datasets, while being on par for the others. Since the chosen datasets not only use medical question answering, but also vision problem settings, this is a strong result.
- Sufficiently deep discussion of results: The authors discuss the results in sufficient depth, highlighting reasons for the added value of using the novel multi-agent setup, especially categorizing the severity of cases by an initial agent.

**Weaknesses:**

- The paper could me more specific on method descriptions, e.g., multi-party chat / collaboration / report generation: The description is quite generic in the main paper and one only finds examples in the appendix. It is left open for me what implications the unknown design decisions have on the performance of the system. The same holds for, e.g., how the optimal number of collaborating agents is chosen on a case basis.
- The paper only uses 50 samples for inference without argumentation: It might be fine if other works to the same, but it should be clearly stated why 50 samples where chosen. Maybe the close competitors, e.g., MedAgents, do the same? What percentage is this for the respective datasets?
- Possible missing details on the evaluation setup: It might be that it is obvious or simply not used, but I am unsure when and where zero-shot or few-shot prompting is used. I see it for the low complexity cases for the recuriter LLM if the approach and also for solo competitors in the evaluation. Is it not used at other places? Extending on this point, maybe it would be helpful to emphasize such a point in the paper or point to it in the appendix for all used, established prompting strategies. If none is used in additition to the role descriptions and collaboratin schemes, it might be also worth noting.

**Questions:**

- Please explain why, for the evaluation, you use 50 samples for inference and why it suffices / not have an impact on the relative results to the competitors.

- Please better explain, also in the paper, what a multi-party chat (mentioned in table 1) would look like for specific tasks. The case studies with MDAgents in the appending is insightful, but a better formalization/presentation of the topic in the main paper would be helpful.

- Are the expert discussion always one-to-one? If so, will there always be exhaustive one-to-ones?

- How dynamic is the system with respect to generating teams of experts of high severity cases? The meta analyses show that more agents are not necessarily better and that the system calibrates the number of collaborators, but how is it done in detail?

- Are the individual expert LLMs also variated (in terms of prompts)?

- Why was MedVidQA evaluated with Gemini-Pro(Vision)?

- To be sure my understanding is correct: are the individual LLMs mostly zero-shot learners? For few-shot learners (maybe only in the low complexity setting): how many dataset-specific examples, if any, are used for prompting in the evaluation?

### After author response ###
I appreciate the detailed answers. After additionally taking into account the answers and other reviews, I increase my score to accept, as the claims are now clearer backed up.

**Limitations:**

The authors discuss limitations to sufficient degree, including the lack of comparison to human clinicians or, most importantly, the danger of using systems that possibly hallucinate in critical situations.

---

> ### Author Rebuttal · Authors · 2024-08-07
>
> We appreciate your thoughtful review and the recognition of our paper’s potential contributions to the field. Your insights are invaluable in guiding us to enhance our work and clarify the findings.
>
> **W1. & Q2. Lack of detailed descriptions and examples**
>
> We recognize the importance of providing more detailed descriptions to clarify our methods, particularly in the areas of multi-party chat, collaboration, and report generation. In the revised version, we will enhance our explanations and include visual aids for better understanding.
>
> * **Number of Agents:** We will explain in Section 3.4 how the recruiter LLM determines the optimal number of agents, considering task complexity and expertise, as outlined in Appendix C.2.
>
> * **Report Generation:** We will expand on the report generation process in Section 3.4 by including the specific prompt used and detailing how information is synthesized from multiple agents, building on the example in Figure 12.
>
> * **Multi-Party Chat and Collaboration:** We will provide a more comprehensive description of agent interactions and collaboration dynamics in the main text, focusing on their role in fostering effective teamwork, as illustrated in Figure 11.
>
> **W2. & Q1. Why use only 50 samples in the experiments?**
>
> We selected 50 samples to maintain consistency with similar studies, such as [1], which employ a sample size 50 for evaluation. This decision was made to ensure a manageable yet statistically significant analysis with three random seeds, aligning with our experiment budget constraints (experiments cost listed in Table 3 in attached pdf file).
>
> In our experiments, we utilized the gpt-4 (vision) model, which is approximately up to 200 times more expensive than the gpt-4o-mini model. Given this consideration, we increased the number of samples to 100 with gpt-4o-mini and reproduced our experimental results to provide a more comprehensive evaluation (Table 1 in attached pdf file).
>
> In particular, for MedQA, we evaluated the entire test set to demonstrate that performance trends remain consistent across different models, thus reinforcing the reliability of our findings (Table 2 in attached pdf file). We believe this approach balances practical constraints with the need for rigorous evaluation and hope this addresses your concerns.
>
> Additionally, it is important to note that our use of multiple datasets reflects our intention to test the multi-agent system across various medical scenarios, acknowledging the diversity and complexity of medical diagnostics. By selecting a smaller number of samples from each of the ten datasets, we aim to capture a broad spectrum of medical conditions and scenarios, making our study comparative and comprehensive relative to others..
>
> Lastly, the percentage that these 100 samples represent of the total test set sizes varies, which can be calculated based on Table 5 in our Appendix:
>
> * MedQA: 0.0078%
> * PubMedQA: 20%
> * DDXPlus: 0.074%
> * SymCat: 0.027%
> * JAMA: 6.56%
> * MedBullets: 32.47%
> * MIMIC-CXR: 6.53%
> * PMC-VQA: 200%
> * Path-VQA: 0.00295%
> * MedVidQA: 64.52%
>
> **W3. & Q7. Details missing in evaluation setup and prompting strategies**
>
> In our framework, we employed a 3-shot setting for low-complexity cases where PCP agents make decisions. For moderate and high-complexity cases involving multi-LLM agents, we used a zero-shot setting. We will update our experimental setup in the revised paper to clearly indicate where zero-shot and few-shot prompting are used, and we will add this notation to the relevant figures.
>
> **Q3. Is expert discussion one-to-one?**
>
> In our implementation, we allowed the LLMs to decide if they wanted to engage with other agents. Rather than limiting interactions to one-to-one, we facilitated many-to-many discussions, where multiple agents could interact simultaneously. We will add more detailed descriptions in Section 3.4 and clarify this aspect in Figure 2.
>
> **Q4. How does the system dynamically calibrate expert teams for high severity cases?**
>
> The system dynamically calibrates expert teams based on the complexity of the medical query. Initially, the recruiter LLM determines the number of agents allocated to each team. For the meta-analysis, we fixed the number of agents to assess the impact on performance, rather than allowing the recruiter LLM to decide dynamically. In our implementation, instead of removing the agents during discussions,  we ask each LLM agent to participate actively by contributing when they have relevant insights or corrections.
>
> For moderate complexity case, agents could engage in discussions to clarify or emphasize points in each round / turn. In high complexity case, a lead clinician agent guided the process, asking assistant LLMs for specific investigations and deciding which agents to consult based on their expertise.
>
> **Q5. Are Individual Expert LLMs Variated by Prompts?**
>
> Yes, we assigned specific roles to the LLMs with detailed descriptions during the initialization step. These roles were determined by the recruiter LLM, to give LLM a clear function, which allowed for varied interactions based on the prompts. The initialization prompt is detailed in Appendix C.2, where we show the Agent initialization prompt.
>
> **Q6. Why was MedVidQA evaluated with Gemini Pro Vision?**
>
> We evaluated MedVidQA with Gemini-Pro Vision because it was the only model capable of effectively handling both text and video inputs with reasonable performance. Alternatively, we can sample frames from the videos and transcribe (e.g. whisper) the spoken text to provide images and text to vision LLMs (e.g. gpt-4v), but Gemini-Pro Vision offered a more integrated solution for video input.
>
> We hope our responses have addressed your concerns. Please do not hesitate to let us know if any further explanations or updates are needed to assist in the re-evaluation of our paper.
>
> **Reference**
>
> [1] Nori et al. (2023). Can generalist foundation models outcompete special-purpose tuning? case study in medicine.

---

> > ### Comment · Reviewer_g9qW · 2024-08-12
> > **Thank you for the updates**
> >
> > I appreciate the authors' answers to my questions and provided updates, and have no additional questions right now.

---

### Author Rebuttal · Authors · 2024-08-07

We would like to thank all reviewers for their valuable and constructive feedback on our submission. We are encouraged that they found the work to be important and novel with a comprehensive evaluation and strong results.

Based on the reviews we have made significant updates to our paper and would like to highlight these changes:

**1. Additional experiments with an increased number of samples.**

We have conducted additional experiments by increasing the sample size from 50 to 100 for all benchmarks. We were initially limited by the computational and economic cost of running a larger sample, we do believe that 100 samples provide a robust performance number. To support this, we have performed an evaluation on the entire test set for the MedQA dataset. The results from these expanded experiments, detailed in the attached pdf file, demonstrate that the performance trends are consistent across all benchmarks. This consistency suggests that our initial selection of 50 samples and the new results with 100 samples are representative of the performance on the whole dataset.

To reiterate, the decision to use 50 samples initially was based on our intention to align with previous studies, such as [1], which employed an identical sample size in their initial version of the paper. Moreover, our choice was influenced by the constraints of our experimental budget (refer to Table 3 in attached pdf file), particularly the computational costs associated with using the gpt-4-turbo model.

We also plan to further add experimental results incorporating 100 samples with 3 random seeds in the camera-ready version to ensure meaningful comparisons and enhance the robustness of our findings.

**2. Medical complexity annotations obtained from human physicians**

We have conducted an annotation study with three physicians to evaluate the complexity of 50 representative questions from the MedQA dataset. The questions were carefully selected to ensure they required equivalent medical expertise across different USMLE steps (1, 2, and 3), providing a comprehensive assessment of complexity.

Two among the three physicians had two years of medical training Internal Medicine (Post graduate year 2 (PGY-2) and one among them is a general physician. They rated the questions on a scale of -1 (low), 0 (moderate), and 1 (high). To assess inter-rater reliability, we calculated the Intraclass Correlation Coefficients (ICC), focusing on the most informative types:

* ICC2k (Two-way random effects, average measures): 0.269 [-0.14, 0.55]

* ICC3k (Two-way mixed effects, average measures): 0.280 [-0.15, 0.57]

These ICC values indicate moderate agreement among the raters, highlighting the inherent complexity and subjectivity in evaluating medical questions. The variability in ratings could be attributed to differences in individual experience, interpretation of the question's context, and the nuances of medical knowledge.

Additionally, we used several LLMs to annotate the same questions and compared their assessments with the majority opinion of the physicians. To determine the majority opinion among the physicians, we calculated the mode of their ratings. If the ratings were entirely different (e.g., -1, 0, 1), we used the mean value as the final complexity scale, ensuring a balanced representation when no consensus was reached.

* gpt-4o-mini: -0.090 correlation

* gpt-4o: 0.022 correlation

* gpt-4: 0.070 correlation

* gemini-1.5-flash: 0.110 correlation

The LLMs showed low agreement with human complexity judgments, reflecting the challenges in automating nuanced medical assessments. Differences in physician ratings show subjectivity, suggesting that clear guidelines could improve consistency. We believe enhancing LLMs with better context understanding, medical knowledge, and diverse training datasets may help improve alignment with human physicians.

**Reference**

[1] Nori, H., Lee, Y. T., Zhang, S., Carignan, D., Edgar, R., Fusi, N., ... & Horvitz, E. (2023). Can generalist foundation models outcompete special-purpose tuning? case study in medicine. arXiv preprint arXiv:2311.16452.

---

### Decision · Program_Chairs · 2024-09-25

**Decision:**

Accept (oral)

**Comment:**

In this manuscript, the authors propose MDAgent, a multi-agent framework based on LLMs for medical decision making. The main novelty is a crafted collaboration scheme of multiple agents with designated roles, where a medical question is categorized into being of low, medium or high complexity. Based on the complexity checking result, MDAgents assigns a single primary care clinician LLM agent (for low complexity), a team of multidisciplinary LLM agents (for moderate complexity), or a team of integrated care LLM agent (for high complexity). The agent collaboration adopts multi-turn discussion and iterative report refinement. As much the question is complex, greater will be the benefit of multi-agent approach.

The authors during the rebuttal provided satisfactory answers concerning the design of the empirical evaluation. The critical points were the extension of test sample (50 questions versus 100) and how realistic is the simulation of the clinical decision-making scenario (single-turn versus multi-turn interactions). They shared additional results, according to the request of the reviewer, as a proof of robustness for the property of the proposed model. The Reviewer recognized and acknowledged the additional effort of Authors in answering the questions.

This work indicates an interesting further research directions as follow-up of the empirical investigation: how to modulate the expertize of the team of agents in relation to the heterogeneity of expertise of physicians. The issue is how to balance the agreement/disagreement among physicians and the agreement/disagreement among the agents.

Despite this work presents ethical issues related to the critical health domain, e.g. hallucinations in complex situations, I believe that this contribution should be evaluated from the computational point of view and not from the perspective of deployment or provisioning as a service.